# Patrilineal segmentary systems provide a peaceful explanation for the post-Neolithic Y-chromosome bottleneck

Léa Guyon [1] ✉, Jérémy Guez[1,2], Bruno Toupance [1,3], Evelyne Heyer[1] & Raphaëlle Chaix [1] ✉

Studies have found a pronounced decline in male effective population sizes worldwide around 3000–5000 years ago. This bottleneck was not observed for female effective population sizes, which continued to increase over time. Until now, this remarkable genetic pattern was interpreted as the result of an ancient structuring of human populations into patrilineal groups (gathering closely related males) violently competing with each other. In this scenario, violence is responsible for the repeated extinctions of patrilineal groups, leading to a significant reduction in male effective population size. Here, we propose an alternative hypothesis by modelling a segmentary patrilineal system based on anthropological literature. We show that variance in reproductive success between patrilineal groups, combined with lineal fission (i.e., the splitting of a group into two new groups of patrilineally related individuals), can lead to a substantial reduction in the male effective population size without resorting to the violence hypothesis. Thus, a peaceful explanation involving ancient changes in social structures, linked to global changes in subsistence systems, may be sufficient to explain the reported decline in Y-chromosome diversity.

By analysing mitochondrial and Y chromosome sequences in more than 300 contemporary human populations worldwide, Karmin et al.[1] highlighted that the male effective population size of these populations (estimated from the paternally transmitted Y chromosome) underwent a severe bottleneck around 5000 years ago. This remarkable decrease was not observed for the female effective population size estimated from the maternally transmitted mitochondrial DNA (mtDNA). These results were further supported by a study on 26 human populations from the 1000 Genomes Project, which found evidence for a similar bottleneck of male effective population size[2]. Similarly, Batini et al. analysed genetic diversity from 17 populations from Europe and the Near East, and showed similar Y chromosome pattern[3,4]. The estimated timeframe of this bottleneck varies between world regions, ranging from 8300 BP in the Near East to 1400 BP in

Siberia, and was estimated to 5000 BP in Europe according to Karmin et al.[1] using a mutation rate of $0.74 \times 10^{-9}$ mutations/bp/year for the Y chromosome. Using a slightly higher mutation rate ($1.0 \times 10^{-9}$ mutations/bp/year), Batini et al.[3] estimated more recent dates for this bottleneck (ranging from 4200 to 2100 BP for the Near Eastern and European populations)[3].

Several authors have pointed out that the interpretation of patterns in human genetic diversity needs to take into account fine-scale socio-cultural factors highlighted by archaeological and anthropological studies[5–8]. The first socio-cultural model proposed to explain the Y chromosome bottleneck involved violent competition between patrilineal kin groups and the killing of large numbers of men[9]. Thus, Zeng et al.[9] simulated a human population structured into competing patrilineal kin groups. In this model, kin groups−referred to as descent

[1]Eco-Anthropologie (UMR 7206), Muséum National d'Histoire Naturelle, CNRS, Université Paris Cité, Paris 75116, France. [2]Université Paris-Saclay, CNRS, INRIA, Laboratoire Interdisciplinaire des Sciences du Numérique, Orsay 91400, France. [3]Université Paris Cité, Eco-anthropologie, Paris F-75006, France. ✉e-mail: lea.guyon@edu.mnhn.fr; raphaelle.chaix@mnhn.fr

groups in the present article, following Fox's terminology[10]—are genetically homogeneous for the Y-chromosome because men in the same group share a recent common paternal ancestor. Violent competition between these different patrilineal groups results in the extinction of certain groups. By increasing the rate of loss of Y-chromosomal lineages and accelerating genetic drift, it induces a severe decrease of Y-chromosome diversity.

However, it is questionable whether violence is necessary to explain the genetic signal observed by Karmin et al.[1] as the level of violence worldwide at these times remains uncertain (Zeng et al.'s model assumes that 15% of males were killed at each generation[9]). Using ethnographic and/or archaeological data, some authors have claimed that Pleistocene hunter-gatherers exhibited low levels of warfare[11,12], while others came to the opposite conclusion[13–15], fuelling the long-standing debate between Rousseauians and Hobbesians about human nature[16]. Furthermore, a study based on data from various archaeological sites in the Old World found that levels of violence remained stable from the time *Homo sapiens* diverged from other primates until the Bronze Age, but significantly increased during the Iron and the Middle Ages[17]. On the other hand, other studies have reported an increase in violence in more ancient periods, such as during the Early Neolithic in Central Europe[18], and during the Copper Age and Late Bronze Age in the Middle East[19]. These inconsistencies likely result from the nature of the archaeological record, which suffers from numerous biases, including the particularly small sample sizes for the earliest periods. Moreover, although a large number of individuals have been found buried with weapons in the archaeological records of the Bronze Age, this does not coincide with an increase in interpersonal violence in either Europe or the Middle East, suggesting that the idealisation of the social persona of the warrior may not reflect an intensification of warfare[19–21].

Previous studies have demonstrated that kinship systems, and particularly residence and descent rules, leave on their own a profound signature on uniparental genetic diversity without any need for violent competition[22,23]. For instance, patrilocality, a residence rule whereby a married couple settles in the house of the husband or his father, has been shown to reduce local Y-chromosome diversity compared to mtDNA, because men remain in their native village while women migrate between villages with their mtDNA upon marriage. This pattern was reported in various populations, such as hill tribes of Northern Thailand[24–26], food-producers of Sub-Saharan Africa[27], tribes of Western New Guinea[28], and Sumatran populations[29]. On the other hand, matrilocality, the rule according to which the couple settles near the wife's place of birth, is known to reduce diversity on mtDNA compared to the Y chromosome within groups (e.g. in the West Timor population[30], Ngazidja of Comoros Islands[31], and South-East Asian populations[32]).

The patrilineal descent rule, according to which individuals are affiliated with the descent group of their father, is also known to reduce Y-chromosome diversity. Indeed, within patrilineal lineages or clans, men are closely related on paternal genealogical lines and share similar Y chromosomes[33–36]. In Central Asia, present-day patrilineal populations were shown to have lower male than female effective population sizes[37], along with reduced Y-chromosome diversity as compared to non-patrilineal (but patrilocal) populations[36,38]. These populations practice group exogamy, resulting in the migration of women between descent groups.

Here we test the hypothesis that the Y-chromosome bottleneck is the result of a global shift towards patrilineal systems, associated with the transition to new subsistence systems on all continents over the past 12,000 years. Supporting this hypothesis, previous work has shown that kinship systems covary with production systems. Marlowe[12] showed that patrilocality is over-represented in contemporary non-forager populations (mainly farmers and herders) compared to forager populations (60% and 34% of populations,

respectively). Similarly, patrilineal descent is over-represented in non-foragers compared to foragers (47% and 14%, respectively). This has been famously summarised by Aberle: 'the cow is the enemy of matriliny'[39]. More recently, Holden and Mace confirmed that such relationships between subsistence and kinship systems exist in the Bantu populations even after correcting for the historical relationships between populations[40]. We can therefore assume that the emergence of agro-pastoralism was accompanied by a change in the kinship system through the practice of patrilineality and patrilocality. Two main hypotheses have been advanced to explain the relationship between subsistence strategies and kinship systems. According to structural functionalists, patrilocality and patrilineality arise when resources can be accumulated. Indeed, with movable property, prosperous men can offer a bride price to the parents of marriageable young women, rather than moving in with their in-laws for bride service. In this way, daughters are separated from their parents, while men remain in their place of birth after marriage. More generally, movable property is thought to empower men to resist matrilineal and matrilocal traditions[41–43]. According to evolutionary anthropologists, patriliny is more prevalent in wealth-accumulating societies because this inheritance rule is better suited to maximising reproductive success due to the higher reproductive potential of males compared to females[40,44].

Two features of patrilineal systems may be particularly relevant for the evolution of uniparental genetic diversity. First, it was observed that in contemporary patrilineal populations not necessarily involved in violent conflict, demographic stochasticity cause some groups to grow over time as the number of descendants of the group founder increases, while other groups may die out by chance[10]. As Fox wrote, 'it is the problem of all societies based on unilineal descent groups that these groups are subject to such fluctuations—some expanding and growing abnormally large, while others decline and become extinct'[10 p.122]. Differential access to resources and differences in social status between groups may reinforce this process, with the high-status groups having greater reproductive success than the low-status groups. This variance in reproductive success between descent groups has been observed, for example, in China, where high-status imperial lineages from the 18th to 19th centuries have higher growth rates and lower extinction rates compared to low-status patrilineages[45]. This has also been observed in north-west Ireland, where about one in five males is likely to be descended from a very powerful and long-lived early medieval dynasty: the *Uí Néill* (see Supplementary Tables 1, 2 for a summary of the variance in growth rates and extinction rates reported for patrilineal lines and groups in the literature). In both cases, reproductive success is transmitted within descent groups, and groups with a high social status tend to have more children in each generation than groups with a low social status. Such variance in reproductive success between descent groups and its intergenerational transmission are expected to accelerate the growth of certain lineages and the extinction of others, at a faster rate than compared to a population with no difference in status.

Secondly, some patrilineal systems, known as segmentary patrilineal systems, are characterised not only by patrilineal descent but also by the segmentation (also called fission) of descent groups when they become too large[10,46]. These systems account for 14% of patrilineal systems according to ref. 39. Fissions are likely to occur in a lineal way, following paternal genealogical lines. In other words, paternally related men cluster together in the newly formed patrilineal groups[47–50] (Supplementary Fig. 1). This type of fission has been shown to reduce genetic diversity within groups and to increase differentiation between groups more severely than random fission, where newly formed descent groups arise randomly without regard to paternal relatedness between men[48]. Although Zeng et al.'s model takes into account the fact that patrilineal descent groups experience fissions, these events are modelled as random fissions and not as lineal fissions[9]. Lineal fission, because it clusters paternally related men together, is expected

to be more efficient than random fission at increasing Y chromosome similarity within groups. It accelerates the loss of Y-chromosome diversity as compared to random fission when groups become extinct.

In this study, we undertake a modelling approach to test our hypothesis that a transition to patrilineal organisations, linked to a worldwide change in subsistence strategies, may have triggered an important loss of Y-chromosome diversity and may be sufficient to explain the post-Neolithic Y-chromosome bottleneck reported by Karmin et al.[1] without requiring a violent scenario. By simulating different socio-demographic models, we assess the effect of patrilocal residence and patrilineal descent on uniparental genetic diversity. We modelled both males and females and calibrated our mutation model for mtDNA and the Y chromosome using mutation rates from the literature, so that the relative levels of diversity in the simulated mtDNA and Y-chromosome data can be compared with those reported in the literature[1,3,4]. We integrated population structure by modelling a population structured into five villages. Fission was modelled as either lineal or random, with the former grouping males closely related on their paternal genealogical lines into new descent groups. We also accounted for the fact that different descent groups may have different growth rates, by calibrating such variance with data from studies of patrilineal lines or groups not mentioning warfare between them. Finally, we considered the possibility that descent groups may migrate to another village after a fission. To monitor the change in male and female effective population sizes in our model, we used two different methods that were previously reported in the literature. The first method, similar to the one used by Zeng et al.[9], is to calculate uniparental genetic diversities at multiple time points. The second method, which was used in Karmin et al.[1], is a Bayesian method based on coalescent, inferring the change in effective population size from contemporary individuals. We show that variance in reproductive success between patrilineal groups, combined with lineal fission, can lead to a substantial reduction in the male effective population size without resorting to the violence hypothesis. Thus, a peaceful explanation involving a shift in ancient social structures in relation with the rise of agro-pastoral subsistence strategies may be sufficient to explain the reported decline in Y-chromosome diversity.

## Results

### Socio-demographic model

In order to evaluate whether a transition towards patrilineal organisation could explain the post-Neolithic Y-chromosome bottleneck reported by Karmin et al. and other studies[1–3], we developed a socio-demographic model initially including 1500 individuals, with males carrying a 1 Mb Y chromosome and females a 10 kb mtDNA. The population size is constant for 20,000 generations and individuals reproduce panmictically. Then, 100 generations before present, the population is divided into 5 villages of equal size and begins to grow exponentially (1% per generation[51–53]). Exponential growth is introduced in the model to mimic the dynamics of female effective population size shown in Karmin et al.[1]. Residence is patrilocal, so females migrate between villages more than males. The female migration rate is set to 10% per female per generation. The male migration rate is either zero or 2% per male per generation. This small percentage of male migration corresponds to cases of male adoption, adultery, or flexibility of the residence rule[32,54].

In scenario 1 (Fig. 1), the descent rule is bilateral (the child is affiliated with the maternal and paternal families, there is no descent group) and in scenario 2, the descent rule is patrilineal (the child is affiliated with the descent group of the father). In the latter case, villages are initially divided into three descent groups of equal initial size. Descent groups practice group exogamy (males mate preferentially with females from other descent groups). Reproduction occurs at the village level after migration of individuals. In the patrilineal case, mating pairs are formed by randomly drawing individuals from two

different descent groups, unless all remaining single individuals are from the same descent group. In this case, they are randomly drawn from the same descent group. While the size of villages increases exponentially with a fixed rate (1%), descent groups have different growth rates (see Eq. (1) in Methods). The growth rates of descent groups depend on their relative fitnesses that are initially drawn from a normal distribution with mean $r = 0.01$ and variance $\sigma^2$ that can be zero (i.e. no variance, all descent groups have the same growth rate), low ($\sigma^2 = 0.05$), intermediate ($\sigma^2 = 0.1$) or high ($\sigma^2 = 0.2$). These values have been chosen so that the growth and extinction rates of descent groups are consistent with those found in the literature focusing on patrilineal lines or groups not mentioning warfare between them (Supplementary Tables 1, 2). Descent groups undergo fission when their size exceeds a predetermined threshold $N_{max}$, that can be 100, 150 or 200 individuals, and if the most recent fission event occurred more than 3 generations ago. At each fission event, we assign new relative fitnesses to the two newly formed descent groups. These new relative fitnesses are drawn in a normal distribution centred on the relative fitness of the splitting group, with a variance equal to 0, 0.05, 0.1 or 0.2 (the variance is set to the value of the relative fitness distribution used when the descent groups were initially formed). The greater the variance $\sigma^2$ of the normal distribution, the greater the variance in reproductive success between groups. After a split, the two newly formed descent groups always stay in their village, or the smallest group always move to another village[55,56]. In this article, we will refer to the latter process as post-fission migration. Lastly, descent groups go extinct when their size reaches zero.

We subdivided scenario 2 into eight variants detailed in Table 1 to independently assess the effect of the variance in reproductive success between descent groups, the effect of taking into account lineal fission instead of random fission, and the effect of incorporating or not violent competition between groups in the model. Violence was modelled so that at each generation 15% of males are killed, with a higher probability of being killed when belonging to a small descent group, as in Zeng et al.'s model[9]. We also assessed the impact of a small percentage of male migration, mimicking the effects of adoption, adultery and flexibility in the residence rule, and the impact of post-fission intervillage migration of the smallest newborn descent groups. The eight variants of scenario 2 were simulated with and without these two types of migration. In addition, we evaluated the effects of different levels of variance in reproductive success between groups and different fission thresholds. Supplementary Table 3 compares the settings of our model to those of Zeng et al.'s model[9].

Furthermore, because it is unlikely that a population transitioned directly from panmixia to a patrilocal and patrilineal kinship system, we simulated scenario 3, involving an initial shift from panmixia to a bilateral system (with different residence rules), followed, 100 generations later, by a transition to a patrilineal system. The choice of bilateral descent for the initial phase was motivated by the fact that contemporary hunter-gatherers exhibit most frequently bilateral descent[12]. Some studies, based on contemporary hunter-gatherers, have argued that ancient hunter-gatherers may have been mostly patrilocal[57], while others have argued that they were more likely to be mostly multilocal[12] (where multilocal residence is defined as a rule under which couples can choose to live near the husband's or the wife's place of birth). Finally, the study of isotopic ratios of ancient hunter-gatherer remains suggests a preferentially matrilocal residential pattern[58]. As there is no consensus on the prevailing residence rule in populations before the Neolithic transition, we modelled three different initial shifts: one from panmixia towards a bilateral and matrilocal system (in which males migrate more than females between villages), one towards a bilateral and multilocal system (in which there is an equal proportion of males and females migrating between villages), and one towards a bilateral and patrilocal system (Supplementary Fig. 2). These initial

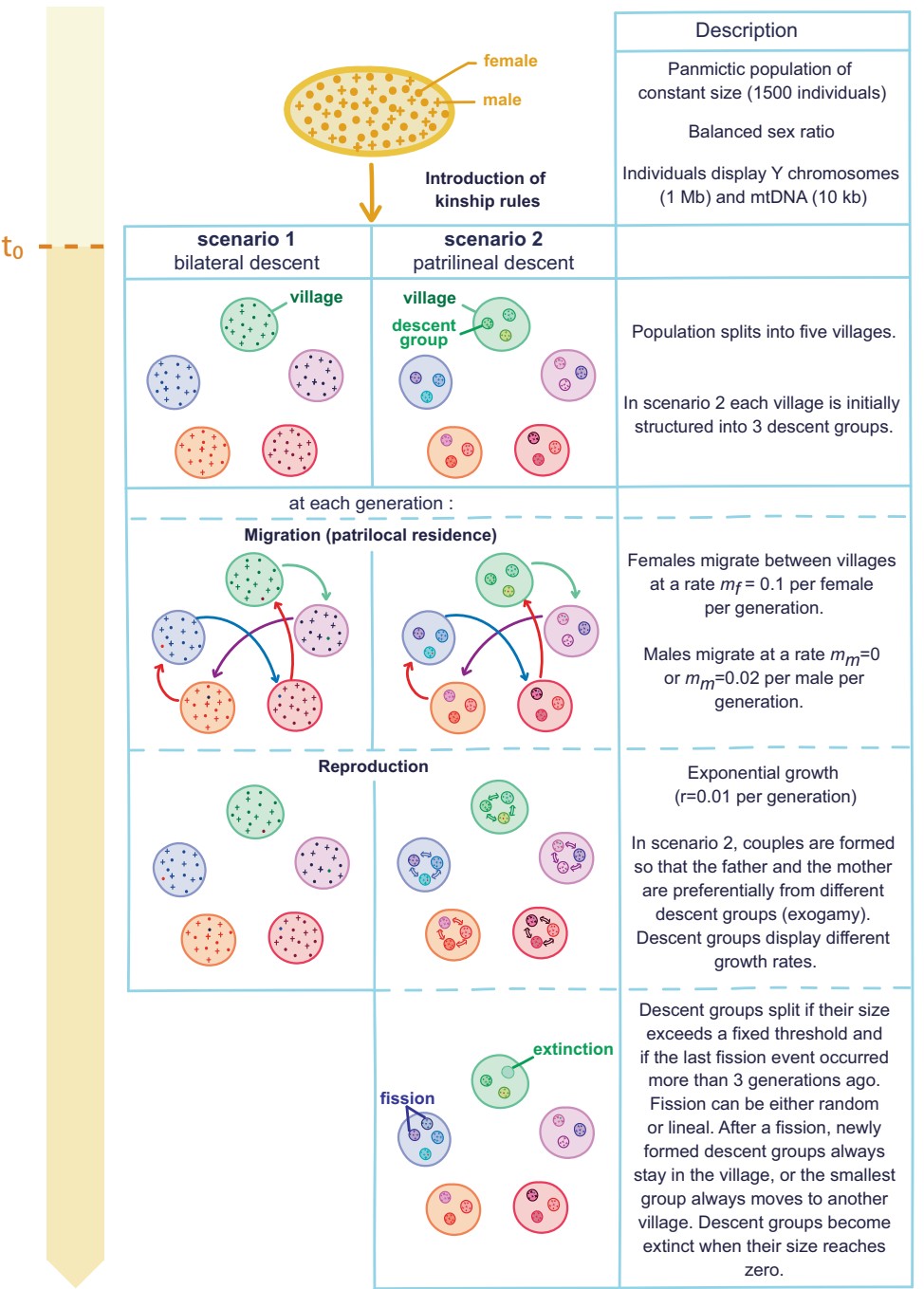

**Fig. 1 | Steps of the simulated scenarios 1 and 2.** At $t_0$ = 100 generations ago (after 20,000 generations of panmixia), the kinship rules of interest are introduced. In scenario 1, the population is divided into 5 villages that follow a patrilocal residence rule, with 10% of females, and either 0 or 2% of males, migrating between villages at each generation. The descent rule is bilateral. In scenario 2, the population is divided into five patrilocal villages, themselves divided into patrilineal descent groups. Mating occurs between individuals from different descent groups (exogamy) unless all remaining mates are from the same group. Patrilineal descent groups can experience fission if their size is over $N_{max}$ individuals and if the last fission event has occurred more than three generations ago. They can also go extinct if they become empty. Cycles of migration/reproduction/fission and extinction are simulated for 100 generations.

shifts are followed 100 generations later by a shift to a patrilineal system.

We used two different methods to estimate the male and female effective population sizes ($N_e$) over time, in order to explore the sensitivity of our conclusions to these inference methods. First, we calculated the average pairwise number of nucleotide differences ($\pi$) over time from the Y chromosome and mtDNA, respectively. This estimate allows us to follow the change in $\pi$-based male and female effective population sizes over time, an approach similar to that used in the study by Zeng et al.[9]. Second, as in the study by Karmin et al.[1], we

generated Bayesian skyline plots (BSPs) showing the change in a coalescent-based estimate of male and female effective population sizes over time using the software BEAST v1.10.5[59].

## $\pi$-based male and female effective population sizes

In the main section of this paper, we present within-village $\pi$-based effective population sizes, as in the study by Zeng et al. which did not integrate population structure[9]. Global $\pi$-based effective population sizes (estimated from individuals across the five villages) are presented in Supplementary information (see Supplementary note 1). In the main

**Table 1 | Description of the simulated scenarios**

| Scenario | Descent rule | Residence rule | Fission type | Variance in reproductive success | Violent competition | Preceded by 100 generations of bilateral descent |
|---|---|---|---|---|---|---|
| 1 | Bilateral | Patrilocal | None | No | No | No |
| 2a | Patrilineal | Patrilocal | Random | No | No | No |
| 2b | Patrilineal | Patrilocal | Random | No | Yes | No |
| 2c | Patrilineal | Patrilocal | Lineal | No | No | No |
| 2d | Patrilineal | Patrilocal | Lineal | No | Yes | No |
| 2e | Patrilineal | Patrilocal | Random | Yes | No | No |
| 2f | Patrilineal | Patrilocal | Random | Yes | Yes | No |
| 2g | Patrilineal | Patrilocal | Lineal | Yes | No | No |
| 2h | Patrilineal | Patrilocal | Lineal | Yes | Yes | No |
| 3a | Patrilineal | Patrilocal | Lineal | Yes | No | Yes (patrilocal) |
| 3b | Patrilineal | Patrilocal | Lineal | Yes | No | Yes (multilocal) |
| 3c | Patrilineal | Patrilocal | Lineal | Yes | No | Yes (matrilocal) |

This table shows the settings for the different simulated scenarios. All scenarios from 1 to 2h have been simulated with or without a small proportion of intervillage male migration (2% per generation), and all scenarios from 2a to 2h have been simulated with or without post-fission group migration. Scenarios 3a to 3c correspond to scenarios with two transitions of kinship rules (see Supplementary Fig. 2).

section, we present results for scenarios without post-fission migration and without 2% intervillage male migration. Results for scenarios with post-fission migration and/or 2% intervillage male migration are presented in Supplementary Tables 4–6 and Supplementary Figs. 3–5.

Table 2 and Fig. 2 show that the male effective population size decreased by the same amount (below 2) after 100 generations when there was a shift to bilateral descent and patrilocal residence (scenario 1) and when there was a shift to patrilineal descent and patrilocal residence with random fission, no variance in reproductive success and no violence (scenario 2a: basic patrilineal scenario) (Fig. 2a). Adding violent intergroup competition (scenario 2b, the most similar to that proposed by Zeng et al.[9]), variance in reproductive success between patrilineal descent groups ($\sigma^2 = 0.1$) (scenario 2e), or both violent intergroup competition and variance in reproductive success (scenario 2f) to the basic patrilineal scenario reduced the male effective population size by 2–4, 100 generations after $t_0$.

Descent groups can undergo lineal fission rather than random fission[50], gathering the most paternally related males into the newly formed groups. Taking into account lineal fission rather than random fission in the basic patrilineal scenario (scenario 2c) did not result in a greater reduction in the male effective population size. However, adding both lineal fission and violent intergroup competition to the basic patrilineal scenario (scenario 2d) reduced the male effective population size by 8, 100 generations after $t_0$. Furthermore, adding lineal fission and variance in reproductive success between groups ($\sigma^2 = 0.1$) to the basic patrilineal scenario reduced the male effective population size by over 20, regardless of the inclusion of violence (2g and 2h), 100 generations after $t_0$. Therefore, lineal fission and variance in reproductive success have the greatest impact on male effective population size.

These results show that the addition of intergroup variance in reproductive success to the model (at levels compatible with those reported by studies focusing on patrilineal lines and groups without mentioning warfare between them), along with lineal fission, is sufficient to induce a substantial reduction in male effective population size, regardless of whether violence is included in the model. We also found that different socio-demographic processes acted at different rates on male effective population size. In particular, violence had a rapid effect on genetic diversity: in scenarios with variance in reproductive success between groups and lineal fission, the decline in male effective population size is much faster in the scenario with violent competition (2h) than in the scenario without violent competition (2g) (Fig. 2a). For example, after 20 generations, the male effective population size decreased by a factor of 3.30 in the scenario 2h (with

violence) and only by 1.36 in the scenario 2g (without violence). However, in the long run, the effect of variance in reproductive success is greater than the effect of violence, and scenarios with violent competition without additional variance are less efficient at reducing male effective population size than scenarios with variance in reproductive success but no violent competition between groups (scenario 2b compared to 2e, and 2d compared to 2g).

Scenarios 2a to 2h were simulated again by considering 2% intervillage male migration and post-fission migration. When considering either or both types of migration, the different scenarios had the same relative effect on male effective population size as described above (without migration), with a greater loss of male effective population size when taking into account variance in reproductive success and lineal fission. Considering 2% intervillage male migration, the reduction in male effective population size was notably less pronounced than in scenarios without such intervillage male migration (Supplementary Table 4, Supplementary Fig. 3). When post-fission migration was considered, the reduction in male effective population size was also less pronounced than in scenarios without this type of migration (Supplementary Tables 5–6, Supplementary Figs. 4–5).

In all scenarios, the female effective population size did not decrease over time, resulting in $\pi$-based female-to-male effective population size ratios similar to the male effective population size reduction factors (Fig. 2, Table 2, Supplementary Figs. 3–5, Supplementary Tables 4–6).

Note that the reduction in male effective population size depends on the variance in reproductive success between groups introduced in the model. As shown in Fig. 3, the male effective population size decreased more severely for higher values of variance $\sigma^2$ of the normal distribution used to draw relative fitnesses of patrilineal descent groups, and the female-to-male effective population size ratio increased more over time, reaching up to 29.42 when $\sigma^2$ is set to 0.2. In other words, the greater the variance in reproductive success between groups, the lower the male effective population size and the greater the female-to-male $N_e$ ratio. On the other hand, the fission threshold had a weak effect on the change in male effective population size for the values we tested (Supplementary Fig. 6).

Taking into account an initial shift to a bilateral system (with different residence rules) lasting 100 generations before the transition to a patrilineal system (also lasting 100 generations) with the settings of scenario 2g (i.e. with lineal fission, variance in reproductive success between descent groups, and no violent competition), we still observed a notable reduction in male effective population size (Fig. 4). In addition, the residence rule prior to the introduction of patrilineality

**Table 2 | $\pi$-based and coalescent-based male effective population size reduction factor and maximum ratio of female-to-male effective population size under the different scenarios**

| Scenario/ study | Male effective population size reduction factor | | Maximum female-to-male effective population size ratio | |
|---|---|---|---|---|
| | $\pi$-based [b] | coalescent-based [c] | $\pi$-based | coalescent-based |
| Karmin et al.[1] [a] | NA | Siberia: 2.4 ; Andes: 2.5 South-East and East Asia: 2.7 South and Central Asia: 2.9 Near East: 3.7; Africa: 4 Europe: 6.7; 3.5 on average | NA | 17 |
| 1 | 1.31 | 0.81 | 1.29 | 1.29 |
| 2a | 1.32 | 0.82 | 1.26 | 1.38 |
| 2b | 2.14 | 1.14 | 2.18 | 2.13 |
| 2c | 1.31 | 0.83 | 1.29 | 1.24 |
| 2d | 8.03 | 2.03 | 7.83 | 4.12 |
| 2e | 3.70 | 1.57 | 3.79 | 3.70 |
| 2f | 2.74 | 1.33 | 2.83 | 3.23 |
| 2g | 21.10 | 2.77 | 21.52 | 7.63 |
| 2h | 24.77 | 3.09 | 25.13 | 8.67 |
| 3a | 20.18 | 2.74 | 22.16 | 19.89 |
| 3b | 20.02 | 2.90 | 21.94 | 18.81 |
| 3c | 19.91 | 2.86 | 22.28 | 18.00 |

For scenarios 1 to 2h, patrilocal residence and the descent rule of interest are introduced at $t_0$, 100 generations before present, after a phase of panmixia. Unless specified otherwise, tables in this paper present results considering a fission threshold of $N_{max} = 150$ (when descent is patrilineal), a $\sigma^2$ parameter controlling the variance in reproductive success between descent groups of 0.1 (in scenarios with variance), a male migration rate between villages set to 0 and no post-fission migration. For scenarios 3a to 3c, the descent rule is bilateral, and the residence rule is either patrilocal, matrilocal, or multilocal, between $t_0$ and $t_1$. Then, at $t_1$, there is a transition to patrilineal descent with patrilocal residence (with the settings of scenario 2g, i.e. lineal fission, variance in reproductive success between groups, no violence).

[a]The reported values were obtained from visual inspection of Karmin et al.'s Fig. S4B1.

[b]Calculated by dividing the number of males in the simulation at $t_0$ (i.e. 750) by the mean $\pi$-based male effective population size, 100 generations after $t_0$ for scenarios 1 to 2h, and 100 generations after $t_1$ for scenarios 3a to 3c.

[c]Calculated by dividing the number of males in the simulation at $t_0$ (i.e. 750) by the minimum coalescent-based male effective population size.

(between $t_0$ and $t_1$) had a weak effect on the reduction in male effective population size after the introduction of patrilineality at $t_1$ since they resulted in similar reduction factors, around 20, after 100 generations of patrilineality.

As shown in Supplementary Tables 7–10 and Supplementary Figs. 7–11, and discussed in Supplementary Note 1, global $\pi$-based male effective population sizes (estimated from individuals across all five villages) are less affected by the different scenarios we tested than within-village $\pi$-based effective population sizes.

**Coalescent-based estimation of male and female effective population sizes**

The bottleneck identified by Karmin et al.[1] on the Y chromosome was obtained not by measuring the evolution of genetic diversity over time, but by inferring past effective population sizes from current genetic data using a coalescent-based method implemented in BEAST[60]. To investigate whether this coalescent-based method yields different results than the effective population sizes derived from the mean number of pairwise differences ($\pi$), we used BEAST to generate BSPs from our simulated Y chromosome and mtDNA data. Unlike the effective population sizes based on $\pi$, those obtained with BEAST have not been computed locally within each village but globally (by sampling all five villages) to match with Karmin et al.'s approach[1], that

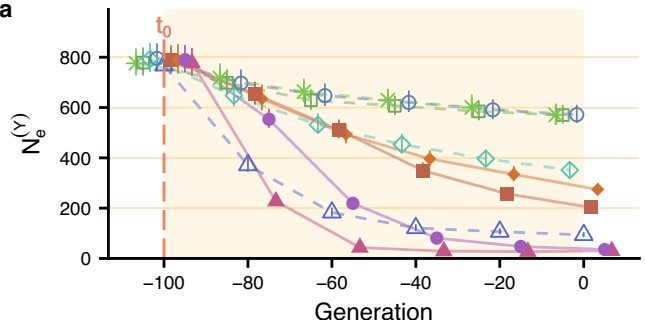

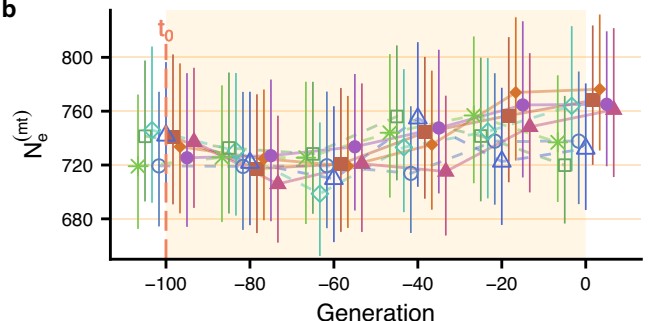

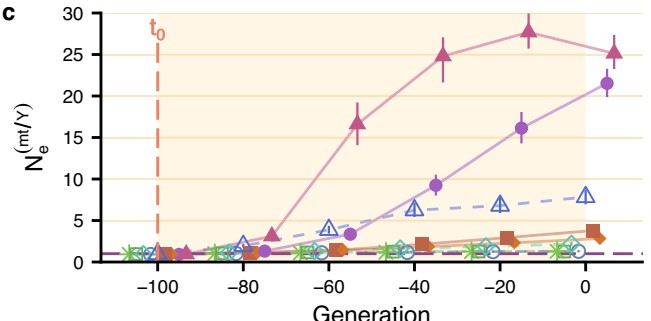

**Scenario**

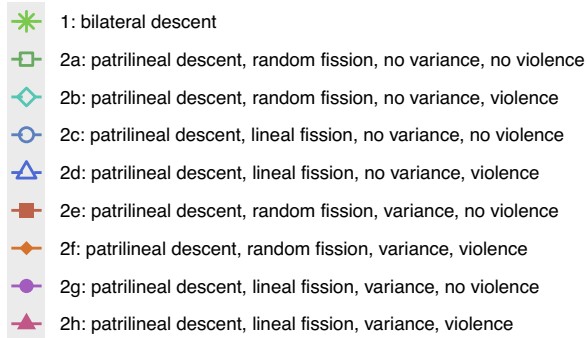

* 1: bilateral descent
* 2a: patrilineal descent, random fission, no variance, no violence
* 2b: patrilineal descent, random fission, no variance, violence
* 2c: patrilineal descent, lineal fission, no variance, no violence
* 2d: patrilineal descent, lineal fission, no variance, violence
* 2e: patrilineal descent, random fission, variance, no violence
* 2f: patrilineal descent, random fission, variance, violence
* 2g: patrilineal descent, lineal fission, variance, no violence
* 2h: patrilineal descent, lineal fission, variance, violence

**Fig. 2 | Change in $\pi$-based male and female effective population sizes over time.** Patrilocal residence and the descent rule of interest are introduced at $t_0$, 100 generations before present, after a phase of panmixia. Unless specified otherwise, figures presented in this paper have been plotted considering a fission threshold of $N_{max} = 150$ (when descent is patrilineal), a $\sigma^2$ parameter controlling the variance in reproductive success between descent groups of 0.1 (in scenarios with variance), a male migration rate between villages set to 0 and no post fission migration. $\pi$-based male effective population size (**a**), female effective population size (**b**), and female-to-male effective population size ratio (**c**) averaged over 200 replicates are shown every 20 generations with 95% confidence interval. In all figures of this paper, the points were staggered along the x-axis to distinguish between scenarios. Values of means and their 95% confidence intervals are provided in the Source Data file.

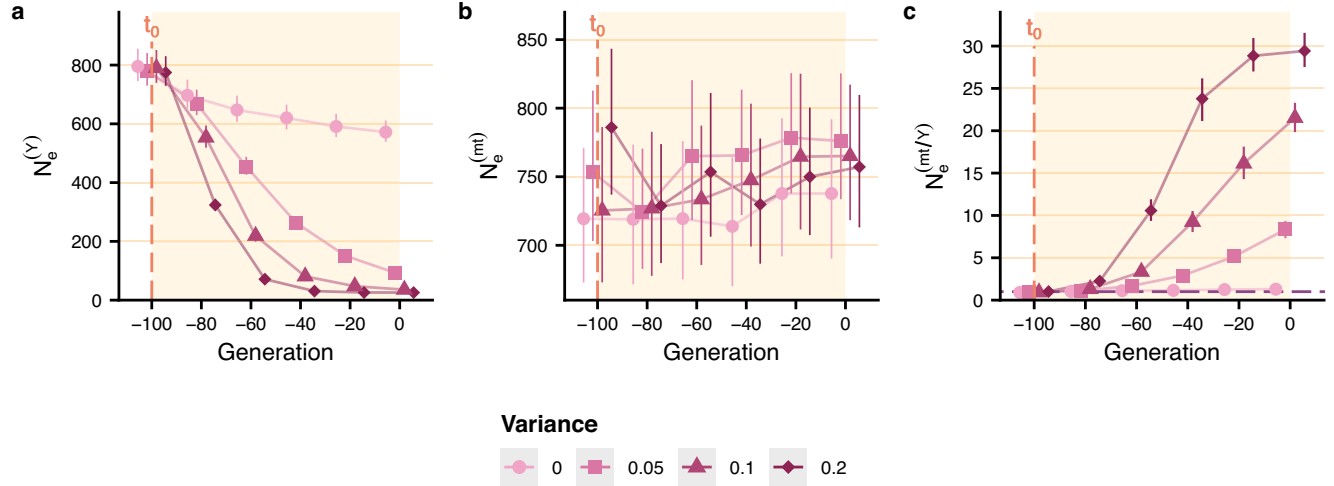

**Fig. 3 | Effect of variance in reproductive success between descent groups ($\sigma^2$) on the change in $\pi$-based male and female effective population sizes.** Patrilocal residence and patrilineal descent (with settings of scenario 2g, i.e. lineal fission, variance in reproductive success between descent groups and no violence) are introduced at $t_0$, 100 generations before present, after a phase of panmixia. See the legend of Fig. 2 for parameter values (except for the parameter controlling the variance in reproductive success between descent groups, which varies between 0 and 0.2). $\pi$-based male effective population size (**a**), female effective population size (**b**), and female-to-male effective population size ratio (**c**) averaged over 200 replicates are shown every 20 generations with 95% confidence interval. Values of means and their 95% confidence intervals are provided in the Source Data file.

computed male and female effective population sizes over several populations.

Figure 5a displays the change in male effective population size over time for each simulated scenario, without intervillage male migration and without post-fission migration (see Supplementary Tables 4–6 and Supplementary Figs. 12–14 for the results with these two types of migration). At $t_0$, all scenarios show different values for the male effective population size. Male effective population size is above 1000 in scenarios with bilateral descent (scenario 1) or with patrilineal descent when there is no variance in reproductive success between descent groups, and no violence (scenarios 2a and 2c). Adding variance in reproductive success between descent groups and/or violent competition resulted in male effective population sizes at $t_0$ between 680 (scenario 2b) and 424 (scenario 2d). Note that these values are either above or below 750, the number of males in the simulations at $t_0$.

For all scenarios, the final male effective population size exceeds the inferred effective population size at $t_0$. However, scenarios 2d to 2h also show that prior to increasing, the male effective population size decreases between −100 and −25 generations. These bottlenecks are seen in patrilineal scenarios with random fission and variance in reproductive success (scenario 2e), or with lineal fission and variance in reproductive success (scenario 2g), or with lineal fission and violence (scenario 2d), or with lineal fission and both variance in reproductive success and violence (scenarios 2f and 2h). However, they do not occur in the scenario with random fission and violence (scenario 2b). We assessed the reduction factors of male effective population size for all scenarios by calculating the ratio of the number of males at $t_0$ (i.e. 750) to the minimum coalescent-based male effective population size (see Table 2). In the scenario with patrilineal descent, variance in reproductive success between groups, and no violence, taking into account lineal rather than random fission, resulted in a greater reduction in the male effective population size (scenarios 2e and 2g in Table 2). In scenarios with patrilineal descent and lineal fission, the introduction of variance in reproductive success (scenario 2g) resulted in a greater reduction in the male effective population size compared to the introduction of violence (scenario 2d). Therefore, variance in reproductive success at levels compatible with populations without mention of warfare between descent groups, combined with lineal fission, is more efficient at reducing the male effective population size

than violent competition. These reduction factors are lower than the reduction factors reached using $\pi$-based effective population sizes. Note however that relatively, the values of both approaches are consistent, with higher reduction factors for scenarios with patrilineal descent, variance in reproductive success between groups, and lineal fission (2g and 2h), and lower reduction factors for scenarios with bilateral descent (1) or patrilineal descent with no variance in reproductive success and no violence (2a and 2c).

Female effective population size increased over time in each scenario (Fig. 5b), resulting in female-to-male $N_e$ ratios ranging from 1.24 (scenario 2c) to 8.67 (scenario 2h) (Table 2) in the scenarios without 2% intervillage male migration and without post-fission migration. These ratios were lower when considering 2% intervillage male migration but no post-fission migration (Supplementary Fig. 12, Supplementary Table 4). However, the ratios increased substantially when post-fission migration was taken into account, with or without 2% intervillage male migration (Supplementary Figs. 13, 14, Supplementary Tables 5, 6). Considering both types of migration, the female-to-male effective population size ratio reached 14.20 in the scenario with patrilineal descent, lineal fission, variance, and no violence (Supplementary Fig. 14, Supplementary Table 6). Note that the average peaks of female-to-male $N_e$ ratios are obtained from the mean BSPs averaged over all replicates. These values are therefore an underestimation of the actual average maximum values of female-to-male $N_e$ ratio of the replicates, as the peaks do not occur simultaneously in the different replicates (Supplementary Fig. 15).

Consistent with the results obtained by measuring nucleotide diversity over time, a higher variance in reproductive success between groups led to a greater reduction in male effective population size, and a higher female-to-male effective population size ratio, reaching up to 9 for $\sigma^2 = 0.2$ (Supplementary Fig. 16). In addition, the value of the fission threshold had little effect on either the reduction in male effective population size or the female-to-male effective population size ratio (Supplementary Fig. 17).

Figure 6 shows that the bottleneck in male effective population size is maintained even when patrilineality is introduced after a phase with a bilateral descent system, regardless of the residence rule. Indeed, the male effective population size reduction factor reached similar values (around 3) with an initial bilateral and patrilocal phase, with an initial bilateral and multilocal phase, and with an initial bilateral

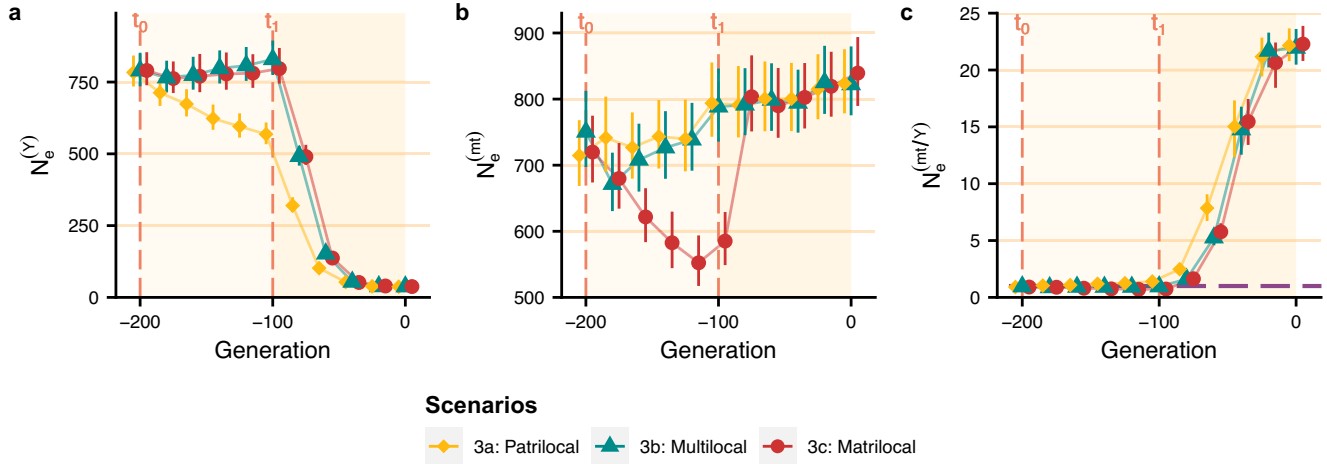

**Fig. 4 | Change in π-based male and female effective population sizes over time under the two transitions scenario.** Between $t_0$ and $t_1$, the descent rule is bilateral, and the residence rule is either patrilocal, matrilocal, or multilocal. Then, at $t_1$, there is a transition to patrilineal descent with patrilocal residence (with the settings of scenario 2g, i.e. lineal fission, variance in reproductive success between groups, no violence, and with the parameter values given in the legend of Fig. 2). π-based male effective population size (**a**), female effective population size (**b**) and female-to-male effective population size ratios (**c**) averaged over 200 replicates are shown every 20 generations with the 95% confidence interval. Values of means and their 95% confidence intervals are provided in the Source Data file.

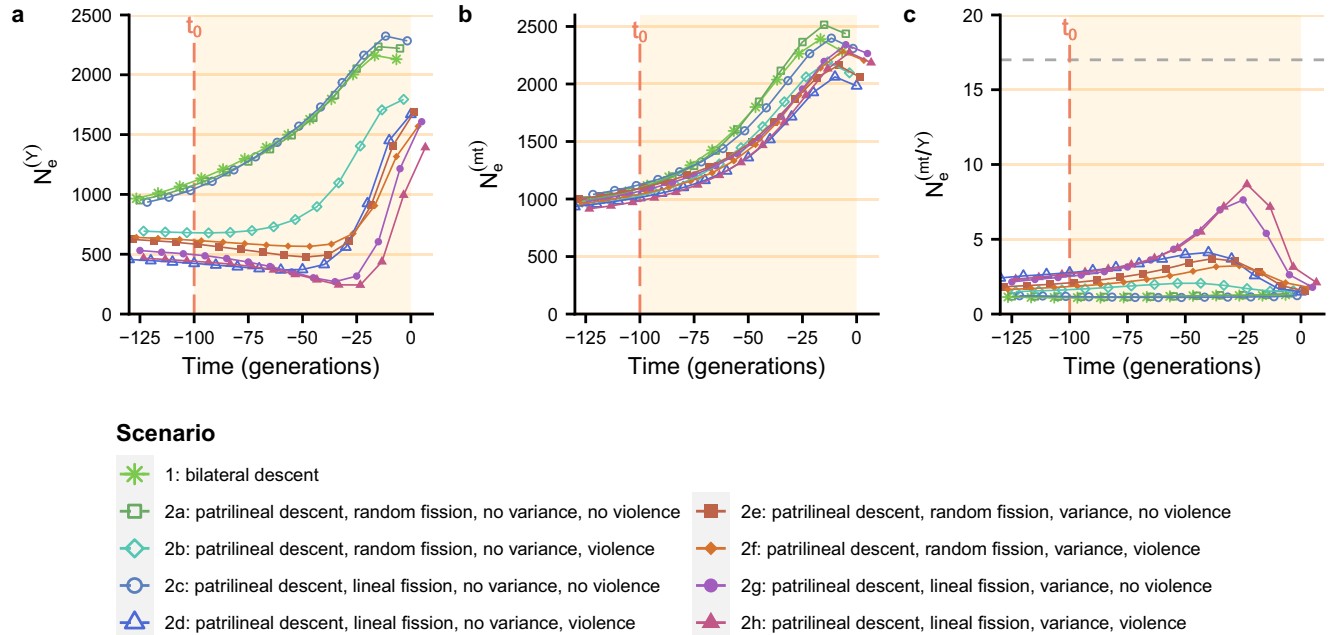

**Fig. 5 | Change in coalescent-based estimation of male and female effective population sizes over time under different scenarios.** Patrilocal residence and the descent rule of interest are introduced at $t_0$, 100 generations before present, after a phase of panmixia. See the legend of Fig. 2 for parameter values. Average Bayesian skyline plots of male effective population size (**a**), female effective population size (**b**), and female-to-male $N_e$ ratio (**c**) over 200 replicates are plotted for each scenario. The dashed grey line corresponds to the ratio of 17 obtained by Karmin et al.[1]. Values of means are provided in the Source Data file.

and matrilocal phase. As the effective population size of females increases earlier than in the previous scenarios, it is higher when the bottleneck in male effective population size occurs. This results in higher peaks in the female-to-male effective population size ratio, which reached values around 19 in all three scenarios.

## Discussion

Previous studies[1–3] reported a substantial reduction in male effective population size worldwide around 3000–5000 BP while the female effective population size was unaffected. To test whether a non-violent scenario may account for this post-Neolithic Y-chromosome

bottleneck, we simulated the evolution of human genomes under different socio-demographic scenarios. We computed male and female effective population sizes over time using the mean number of pairwise differences (π), and a Bayesian inference method. We compared a scenario of patrilocal villages following a bilateral descent rule with scenarios of patrilocal villages structured in patrilineal descent groups with or without variance in reproductive success, undergoing random or lineal fissions, and with or without violent intergroup competition. All these scenarios were simulated with or without a small percentage of male migration (2%) and with or without post-fission migration of the smallest newly formed group after a split.

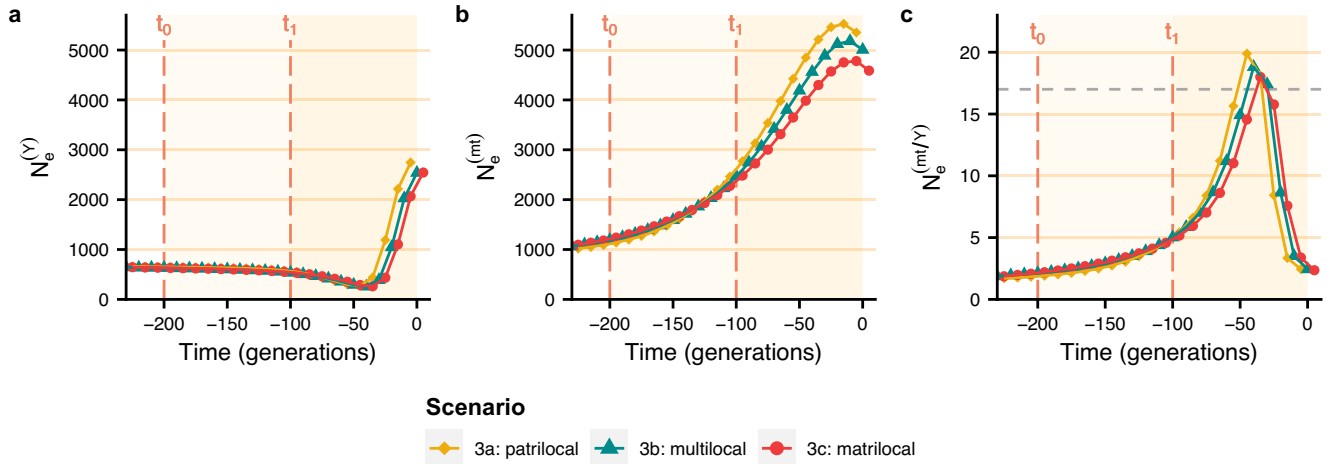

**Scenario**

3a: patrilocal    3b: multilocal    3c: matrilocal

**Fig. 6 | Change in coalescent-based estimation of male and female effective population sizes over time under the two transition scenarios.** Between $t_0$ and $t_1$, the descent rule is bilateral, and the residence rule is either patrilocal, matrilocal, or multilocal. Then, at $t_1$, there is a transition to patrilineal descent with patrilocal residence (with the settings of scenario 2g, i.e. lineal fission, variance in reproductive success between groups, no violence, and with the parameter values given in the legend of Fig. 2). Average Bayesian skyline plots of male effective population size (**a**), female effective population size (**b**), and female-to-male $N_e$ ratio (**c**) over 200 replicates are plotted for each scenario. The dashed grey line corresponds to the ratio of 17 obtained by Karmin et al.[1]. Values of means are provided in the Source Data file.

The two methods used to compute effective population sizes showed similar relative effects of the different parameters in the model but reported different bottleneck amplitudes for the male effective population size. In addition, coalescent-based male effective population sizes were larger after 100 generations of patriliny while $\pi$-based male effective population size were lower, suggesting that the coalescent-based method captures more of the growth signal than the $\pi$-based method. These discrepancies can be explained by the fact that the coalescent-based method infers instantaneous effective population sizes of the population's past from current individuals by measuring coalescence rates along coalescent trees. In contrast, nucleotide diversity ($\pi$) estimates the effective population size of the population by integrating its past dynamics, which is equivalent to computing the average of the coalescence times of all pairs of individuals in the sample.

Our results indicate that the patrilineal segmentary dynamics have a greater impact on the Y chromosome bottleneck than violence alone. In particular, a substantial increase in bottleneck strength was observed when lineal rather than random fission was introduced into the model (Table 2). Fission type combined with variance in reproductive success between groups had a stronger effect on the male effective population size than violence. Lineal fission has been reported in several cases of patrilineal societies[50]. It is more likely to occur in segmentary patrilineal systems[61]. This type of fission is more efficient than random fission at sorting Y chromosomes among groups. Consequently, when a descent group goes extinct, the drop in Y-chromosome diversity is stronger since the Y chromosomes that are lost through extinction are less likely to exist in other descent groups. Our result is consistent with previous theoretical findings showing that lineal fission reduces group effective population size four times more than random fission[48]. Note that without introducing variance in reproductive success between patrilineal groups, there is no difference in the impact of lineal fission versus random fission on Y-chromosome diversity. In both cases, the drop in Y-chromosome diversity is very limited (Table 2). This can be explained by the reduced number of fission and extinction events in scenarios without variance in reproductive success compared to scenarios with variance in reproductive success between groups (around 25 extinction events over 100 generations in scenarios without variance and without violence compared to around 100 extinctions over 100 generations in

scenarios with variance and without violence, see Supplementary Figs. 18–25).

Our observation that, in addition to the effects of lineal fission type, the variance in reproductive success between groups (and its transmission to newly formed groups after a fission event) leads to a marked loss of diversity on the Y chromosome is consistent with previous studies showing that cultural transmission of reproductive success at the individual level leads to a reduction in effective population size[22,62]. Although violence can lead to increased variance in reproductive success between descent groups, it is not the only mechanism contributing to such variance. Indeed, anthropological studies have reported a significant variance in reproductive success among patrilineal lines and groups due to differential access to resources or to a different social status (see Supplementary Tables 1, 2). To calibrate the variance in reproductive success introduced in our model, we computed the mean growth rate of successful descent groups (defined as groups with a positive growth rate) and the mean extinction rate of groups in our simulations and compared them to the growth rates of successful groups and extinction rates measured among patrilineal lines or groups reported in studies that did not mention warfare between them. With an intermediate variance $\sigma^2$ of 0.1, the mean growth rate of successful groups reached 0.14 per generation, which falls within the range of growth rates reported in the literature ([0.12; 2.5] per generation, Supplementary Table 2). Moreover, using the same value for the variance, the mean extinction rate reached 3% per generation, which also falls within the reported range of extinction rates ([0.16; 30]% per generation, Supplementary Table 1). For a discussion on the combined effect of variance in reproductive success and violence, see Supplementary Note 2.

Scenarios considering 2% intervillage male migration and/or post-fission migration showed similar relative results compared to scenarios without these types of migration, with an important effect of variance in reproductive success between groups and lineal fission on the reduction in male effective population size. Surprisingly, adding post-fission migration to the model mitigated the reduction in the $\pi$-based male effective population size (Supplementary Table 5, Supplementary Fig. 4), while it reduced the coalescent-based male effective population size more severely than scenarios without post-fission migration

(Supplementary Table 5, Supplementary Fig. 13). This means that the migration of some descent groups to the other villages accelerates the reduction of coalescent-based male effective population size. Indeed, migrating descent groups experiencing demographic growth may spread their Y-chromosome haplotype throughout the whole population, reducing the overall coalescent-based male effective population size. However, because it introduces new Y-chromosome haplotypes into villages, this post-fission migration reduces the $\pi$-based male effective population size less than when the post-fission migration is not included in the model.

As the transition to patrilocality and patrilineality is unlikely to have occurred directly from a panmictic population, we explored the evolution of genetic diversity under scenarios involving an initial shift from panmixia to a bilateral system (which is the descent system the most frequently observed in contemporary hunter-gatherers[12]), followed, 100 generations later, by a transition to a patrilineal system. The post-marital residence patterns in pre-Neolithic populations are poorly known. It has been argued that they may have had patrilocal residence[63], matrilocal residence[58], or multilocal residence[12]. Consequently, we simulated an initial shift from panmixia to a bilateral system with either patrilocal residence, matrilocal residence, or multilocal residence, followed by a shift to a patrilineal system (Supplementary Fig. 2). All three scenarios showed the same reduction in male effective population size after the introduction of patrilineality. Therefore, a reduction in male effective population size due to patrilineality is compatible with all prior kinship systems. These scenarios also showed that when population growth is older than the onset of patrilineality, the bottleneck of male effective population size occurs at a time when female effective population size is high, leading to coalescent-based female-to-male effective population size ratios around 19, similar to the value of 17 reported by Karmin et al.[1].

In Karmin et al.'s study[1], the male effective population size bottleneck was observed after the Neolithic revolution and was dated at 8300 years BP in the Near East, 5000 years BP in Europe, and 1400 years BP in Siberia, which corresponds to 332, 200 and 56 generations ago, respectively, using a generation time of 25 years. Using a slightly higher mutation rate, Batini et al.[3] dated this bottleneck between 4200 and 2100 BP (i.e. between 140 and 70 generations ago) for Near Eastern and European populations[3]. The BSPs produced from simulated data under the scenario with patrilineal descent, lineal fission, variance in reproductive success, no violence (2g) show that the male effective population size bottleneck reached its maximum around 30 generations before present, 70 generations after the introduction of the patrilocal and patrilineal kinship rules in the population (Fig. 5c). In order to understand which scenario(s) could result in a timing for the bottleneck that is compatible with the timing reported by Karmin et al.[1] and Batini et al.[3], we simulated three additional scenarios. The first scenario (4a) is an extension of the scenario 2g, where patrilineal descent is introduced 200 generations ago (instead of 100). The second scenario (4b) models a first transition from panmixia to a patrilineal and patrilocal system (with the same settings as in scenario 2g), 200 generations ago, followed by a transition to a bilateral and patrilocal system, 100 generations ago. The third scenario (4c) models a first transition from panmixia to a patrilineal and patrilocal system (with the same settings as in scenario 2g), 200 generations ago, followed by a transition to a patrilineal and patrilocal system without variance in reproductive success between groups (as in scenario 2c), 100 generations ago. The BSPs obtained using BEAST are presented in Supplementary Fig. 26. Scenario 4a produced a Y chromosome bottleneck that reached its minimum 40 generations ago (i.e. around 1000 years ago using a generation time of 25 years), while the scenarios 4b and 4c produced a bottleneck around 130 and 110 generations ago (i.e. 3250 and 2750 years ago), respectively, which is more consistent with the timing of bottlenecks inferred from real data by Karmin et al.[1] and Batini et al.[3]. This suggests that for the bottleneck to

have occurred around 3000–5000 years ago, an initial transition to kinship systems that are efficient at reducing Y-chromosome diversity (such as the patrilineal scenario 2g, with lineal fission and variance in reproductive success between groups) must have been followed by a transition to kinship systems that are less efficient at reducing Y-chromosome diversity (here we considered a bilateral descent system or a patrilineal system with no variance in reproductive success between groups).

In the present study, we hypothesise that a global transition towards patrilineal and patrilocal organisations, associated with a transition to agro-pastoralism, explain the Y chromosome bottleneck. In the Near East and Europe, a mixed strategy of animal herding and agriculture emerged in the Neolithic (10,500–6500 BP). It diffused from the Levant into Europe and Iran[64]. In East Africa or the Arabian Peninsula, mobile pastoralism emerged between 7000 and 5000 BP[65]. The reason why the Y-chromosome bottleneck is observed at the end of the Neolithic/ Bronze Age rather than in the early Neolithic may be due to the time lapse needed for patrilineal systems to decrease Y-chromosome diversity, as shown in this study. Alternatively the change in kinship systems may have occurred later, when agro-pastoral economies intensified, which led to higher degree of wealth accumulation and transmission[66]. This may have transformed social organisations towards patrilocality and father-to-son transmission[41,42]. Consistent with the hypothesis of a rapid spread of patrilocality after the Neolithic transition, ancient DNA studies show that male relatedness is often higher than female relatedness within archaeological sites from the Neolithic and the Bronze Age (mostly from Europe). In particular, males tend to share more often the same haplogroup of the Y chromosome, whereas there is a greater diversity of haplogroups for mtDNA[67–77]. In addition, reconstructed pedigrees show more first and second-degree related males than females[67,69,70,72,74–77]. These observations have been interpreted as the result of patrilocal residence (with varying degrees of compliance with the rule), with males more likely to remain in their birthplace and females more likely to migrate between sites. Furthermore, several sites showed evidence of mostly paternal transmission of goods and/or social status (for example refs. 68,69,72,73,75), suggesting patrilineal inheritance systems.

Regarding our finding that segmentary patrilineal systems must have given way to kinship systems that are less efficient at reducing Y-chromosome diversity to explain the timing of the bottleneck, social anthropologists have already suggested that the socio-political importance of descent groups and their structure have probably changed, and/or that these groups were replaced by bilateral kindreds as societies became more politically complex, more centralised, more stratified and/or as they became Christianised, although the exact causal factors as well as the precise nature of the change have remained debated among authors (see for example: refs. 78–80). Future studies will need to investigate the exact nature of this transition.

Other socio-cultural processes may have contributed to the post-Neolithic bottleneck, such as polygyny, which exists in more than 80% of present-day human populations[81]. To assess the effect of high levels of polygyny on the reduction in male effective population size, we implemented an additional scenario by adding polygyny in a bilateral and patrilocal population. Polygyny was modelled following the case of the Kipsigis of Kenya mating system (Supplementary Table 11). The Kipsigis are one of the most polygynous societies in the world[82]. Men have any number of wives as long as they can afford to pay the bride price[83]. Some men have up to 12 wives[84,85] and on average 83% of women have co-wives[86]. In our Kipsigis-like model, males can marry multiple females, where the number of wives per male follows a geometric distribution with a parameter $p = 1/2$. Male effective population size under the Kipsigis-like scenario, did not decrease as much as in patrilineal populations undergoing lineal fissions with violence and/or variance in reproductive success between descent groups ($\pi$-based

and coalescent-based male effective population size reduction factors around 4, Supplementary Figs. 27, 28). This result is consistent with previous studies indicating that polygyny alone is expected to have a reduced effect on genetic diversity[22,87], and would probably not be sufficient on its own to account for the male effective population size bottleneck reported by Karmin et al.[1]. In addition, kinship analyses based on ancient DNA data from the Neolithic and Bronze Age (mostly Europe) showed low frequency of half-brothers and half-sisters[67-71,73-77], with the exception of Hazleton North long cairn[72]. This latter case has been interpreted as evidence for polygyny or serial monogamy. These observations suggest that polygyny was not widespread in those times in Europe. However, cultural transmission of reproductive success from father to son could accelerate the loss of Y-chromosome diversity. In particular, transmitted polygyny has been shown to massively decrease genetic diversity in human populations[22].

Furthermore, an unbalanced sampling in a structured population has been shown to trigger spurious bottlenecks in BSPs[88]. Thus, in a population where males are more structured than females (which is the case in patrilocal and patrilineal populations) if only a small number of demes are sampled, the demographic inference using Y chromosomes is expected to produce a bottleneck, whereas no such signal would be observed using mtDNA. Taking into account such unbalanced sampling could possibly increase the strength of the bottleneck we observed in our simulations.

Finally, a differential effect of natural selection on the Y chromosome and mtDNA could also contribute to increase the female-to-male ratio of effective population sizes. As discussed in Karmin et al.[1], it is unlikely that the male effective population size bottleneck is due to natural selection since this event affected Y chromosome haplotypes from all world regions around the same period 3000–5000 years ago. Previous work has nevertheless shown that natural selection has a greater effect on the Y chromosome than on mtDNA, leading to increased mtDNA over Y-chromosome diversity ratios[89]. Therefore, although selection is unlikely to have caused the Y-chromosome bottleneck by itself, it may have contributed to a reduction in Y-chromosome diversity.

Long-range migrations, such as the movement of Steppe populations from Eastern Europe to Western Europe and Central Asia, occurred around 4500 BP, and have left distinct imprints on contemporary human genomes. Notably, the frequencies of the Y chromosome haplogroups underwent substantial shifts during the Steppe migration, resulting in the overrepresentation of haplogroup R1b1 in Europeans[4,90,91]. Our approach does not formally model these long-range migrations. However, our simulations show that when patrilineal descent is combined with post-fission intervillage group migration, the loss in coalescent-based male effective population size is greater than when this group migration does not occur. Indeed, with group migration, a successful Y-chromosome haplogroup can invade other villages, reducing Y-chromosome diversity. In other words, if Steppe populations were organised into segmentary patrilineal groups as they migrated, this could explain the rapid spread of a particular Y-chromosome haplogroup and the loss of other Y-chromosome haplogroups. Future research should better characterise these interactions between population demography and kinship systems.

In summary, in a context where there is no consensus on past rates of violence in our species, we propose scenarios to explain the post-Neolithic Y chromosome bottleneck that do not assume a high rate of violence. Zeng et al.[9] hypothesised that 15% of males were killed at each generation which is among the highest rates of violence detected during human history[17]. Still, this rate of violence between patrilineal groups had a limited effect on male effective population size and does not allow to explain the 17 to 1 female-to-male effective population size reported by Karmin et al.[1]. Ethnographically calibrated non-violent sources of variance combined with lineal fission had a larger effect on male effective population size, with ratios reaching up

to 19. This shows that violence is neither sufficient nor necessary to explain this bottleneck. Of course, if combined with lineal fission and/or non-violent sources of variance, violence could have contributed to accelerate the loss of male effective population size. The cause of the patterns may therefore be found in between the scenarios proposed by the present study and the study by Zeng et al.[9]. Future research should investigate the relative contribution of patrilineal segmentary dynamics and violent competition to the post-Neolithic Y-chromosome bottleneck. To achieve this, it may be useful to examine the speed of diversity change as the segmentary dynamics and violence impact genetic diversity at different rates. Computations of $\pi$-based effective population size indicate that violence impacts genetic diversity in a faster way. This outcome implies that future research conducting ancient DNA diversity measurements at multiple time points within the same archaeological site may provide insights into the pace of male effective population size reduction, allowing to distinguish between more violent or more peaceful scenarios at local scales and lead to a more comprehensive picture of the socio-demographic processes at work in ancient human populations.

## Methods

### Description of the model

We designed a socio-demographic model using SLiM v 4.1[92], a simulation software that allows the study of the evolution of genetic diversity in forward time under complex scenarios. This software is provided with a set of classes that model entities such as mutations, chromosomes, individuals, and subpopulations. These entities can be simulated under different evolutionary scenarios, including selection events, migration of individuals between subpopulations, specific mate choice, etc.

From the model, we simulated different scenarios summarised in Table 1. Scenarios 1 and 2 involve a shift from a panmictic population to a population structured in 5 villages with patrilocal residence and either bilateral descent (scenario 1) or patrilineal descent (scenario 2) (Fig. 1). Scenario 3 involves a first transition from a panmictic population to a population structured in five villages with bilateral descent, followed by a transition to a population structured in five villages with patrilineal descent. All scenarios were simulated with 200 replicates, which were run in parallel using GNU parallel[93].

**First step: large panmictic population of constant size.** We simulated a panmictic population of constant size $N_{total} = 1500$ individuals with a balanced sex ratio. Males carry a 1 Mb non-recombining portion of the Y chromosome, while females carry mtDNA of size 10 kb.

To ensure complete coalescence in every simulation, it is advised to simulate a burn-in step for more than ten times the total size of the population ($10 \times N_{total} = 15,000$ generations)[92]. Therefore, we simulated a panmictic population of constant size $N_{total}$ for 20,000 generations before introducing the kinship rules of interest.

**Second step: introduction of descent and post-marital residence rules in the population.** After 20,000 generations, the population divides into five villages, each with the same size ($N_0 = 300$ individuals) and a balanced sex ratio. Descent and post-marital residence rules are introduced. In the case of patrilineal descent, each village is structured into three descent groups of initial size $K = 100$ individuals with a balanced sex ratio. The number and size of descent groups may subsequently vary over time as a result of fission and extinction events.

**Migration:** If the residence rule is patrilocal, females migrate more than males between villages. At each generation $t$, a proportion ($m_f = 0.1$) of females moves to a different village. In other words, 10% of females in a village move to a new village at every generation. The male migration rate $m_m$ is either 0 or equal to 0.02 per male per generation. For each migrant, the destination village is chosen at random. If the descent rule is patrilineal, the destination descent group of the

migrant is drawn at random. If the residence rule is matrilocal, a proportion ($m_m = 0.1$) of males, i.e. 10% of males in a village, moves to a different village, at every generation, and the female migration rate $m_f$ is set to 0. If the residence rule is multilocal, a proportion ($m_m = 0.05$) of males, i.e. 5% of males in a village, and a proportion ($m_f = 0.05$) of females, i.e. 5% of females in a village, moves to a different village, at every generation.

**Reproduction**: The population grows exponentially, with a growth rate $r$ set to 0.01 per generation. This growth rate was chosen to be consistent with estimated growth rates during the Neolithic[51–53]. While descent groups may fluctuate in size, the size of villages strictly follows this exponential growth over time, so that all villages are of equal size at any given time. In the case of the bilateral descent rule, offspring result from the reproduction of randomly selected couples. In the case of patrilineal descent, couples are formed by randomly associating a male and a female from two different descent groups within the same village at each generation. Deviations from strict descent group exogamy may occur when a male and a female from different descent groups cannot be mated. In this case, couples are formed by randomly selecting a male and a female from the same descent group, to minimise the number of individuals that remain single. Children belong to the descent group of their father. In absence of polygyny, couples are monogamous. Descent groups display different growth rates. Each initial descent group $i, i \in [1,15]$ is given a relative fitness $r_i$ drawn from a normal distribution of mean $r = 0.01$ and variance $\sigma^2$. At each generation, the offspring population is simulated at the village level. Each child is sequentially assigned to a randomly selected mating pair in the village, such that the probability $P_j$ that the father of the child belongs to the descent group $j$ is given by Eq. (1).

$$P_j = \frac{2 \times M_j \times \exp(r_j)}{\sum_i 2 \times M_i \times \exp(r_i)} \tag{1}$$

where $r_j$ is the relative fitness of the descent group $j$ in the village under consideration, and $M_j$ is the number of males in the descent group $j$.

In the scenario considering polygyny, some males have more than one wife. For each female, a male partner is drawn with replacement. The model mimics the practice of polygyny observed among the Kipsigis of Kenya (Kipsigis-like)[82]. The probability $P_j$ for a male $j$ to be drawn is given by Eq. (2).

$$P_j = \frac{X_j}{\sum_i X_i} \tag{2}$$

where $X_i$ is drawn in a geometric distribution of parameter $p = 1/2$ for each male $i$ of the village.

Supplementary Table 11 shows measures of polygyny for different human populations[82]. We chose to calibrate our polygyny rate on data from the Kipsigis because it is among the populations with the highest reported measures of polygyny[82]. The geometric distribution and its parameter (1/2) were chosen so that in our simulations the ratio of male-to-female variance in reproductive success, corrected by the female-to-male ratio of mean number of children, is similar to that reported for the Kipsigis population (Supplementary Table 11). However, the ratio of the maximum number of children per male to the mean number of children per male in our model is higher than in Kipsigis, which means that we are modelling an even stronger polygyny than what is observed in the Kipsigis.

When children are born, they are alternately assigned a male and a female sex, so as to maintain a balanced sex ratio in the villages.

**Fissions and extinctions**: A descent group becomes extinct when its number of individuals reaches 0. A descent group splits into two new descent groups if twice the number of males exceeds the threshold $N_{max}$ (100, 150 or 200 individuals) and the last fission event occurred more than three generations ago. The minimal time between two fission events was set to three generations so that the mean descent group genealogical depth stabilises around 20 generations (Supplementary Fig. 24), a value that is consistent with descent groups depth in current patrilineal populations[33,94]. If the fission type is random, a new descent group is formed by randomly sampling individuals in the original descent group (Supplementary Fig. 1a). The proportion of males in each newly born descent group is drawn from a truncated normal distribution with mean 0.5 and variance 0.5, lying within the interval [0,1]. If the fission type is lineal, the new descent groups are formed by two groups of males having different most recent common ancestors through the male line (Supplementary Fig. 1b). The proportion of males forming a new group can deviate from 0.5 (similarly as in the random fission case). Once the males are distributed in the two new groups, the females are split between the two groups. For a given group, the proportion of sampled females is equal to the proportion of males sampled from the ancestral group, so that sex-ratio is balanced in the new descent groups.

To account for variance in the transmission of relative fitnesses from the splitting patrilineal group to newly formed groups, new relative fitnesses are assigned to the newly formed groups by drawing them in a normal distribution of mean $r_j$ (the relative fitness of the original splitting group) and variance $\sigma^2$.

**Post-fission migration**: If the model does not account for post-fission migration, newly formed descent groups always remain in their original village after a fission event. If post-fission migration is taken into account, the smallest descent group always moves to another village. The destination village is chosen so that the size of the villages is as balanced as possible. For example, if a fission event occurs simultaneously in two different villages, these villages will exchange one of their newly formed descent groups. In this case, villages where there was not any fission at a given generation will not host any migrating descent group. If there is only one fission event in the whole population, the destination village is chosen randomly.

**Violent competition**: The extinction rate $e$ corresponds to the percentage of males killed due to violence. If there is no violent intergroup competition, $e$ is set to 0. Otherwise, if there is violent intergroup competition, $e$ is set to 0.15, meaning that 15% of males in all villages are killed at each generation. This proportion varies among descent groups. As in Zeng et al.[9], the number of males killed ($D$) in a given descent group $j$ is drawn from a Poisson distribution according to Eq. (3) (with $N_i$ the number of males in group $i$). At each generation, the parameter $\rho$ is scaled so that 15% of males are killed in the population.

$$D \sim Poisson\left( \rho . \left[ \sum_{i:i \neq j} N_i \right] . \sqrt{N_j} \right) \tag{3}$$

Hence, males belonging to small descent groups are more likely to be killed than males belonging to large descent groups. Note that the source of violence is not specified in the model. It could come from local neighbours or from belligerent outsiders. In addition, in the absence of polygyny, violent competition between groups leading to the death of 15% of males at each generation indirectly results in 15% of females remaining single.

All tested parameter values are available in Table 3. Main differences with Zeng et al.'s grid model[9] are reported in Supplementary Table 3.

## Tree management

Every 20 generations, from the introduction of descent and post-marital residence rules, sequences of coalescent trees along the genome are output for each modelled chromosome type[95]. These sequences are composed of only one tree since there is no recombination event in Y chromosome and mtDNA. Tree sequences corresponding to different chromosomes are isolated from each other with

**Table 3 | Tested parameter values**

| Parameter | Range of values |
|---|---|
| Growth rate ($r$) | 0.01 per generation |
| $\sigma^{2a}$ | [0, 0.05, 0.1, 0.2] |
| Polygyny | [yes, no] |
| Male migration rate ($m_m$) | [0, 0.02, 0.05, 0.1] |
| Female migration rate ($m_f$) | [0, 0.05, 0.1] |
| Fission threshold ($N_{max}$) | [100, 150, 200] individuals |
| Fission type | [random, lineal] |
| Post-fission migration | [yes, no] |
| Extinction rate due to violence ($e$) | [0, 0.15] |

[a]Variance of the normal distribution used to draw relative fitnesses of descent groups.

a custom Python script using the package tskit v 0.5.6[96] and pyslim v 1.0.4. Mutations are generated along the coalescent trees with the Python package msprime v 1.3.0[97] using a mutation rate of $2.5 \times 10^{-8}$ mutations/nucleotide/generation for the Y chromosome and of $5.5 \times 10^{-7}$ mutations/nucleotide/generation for mtDNA[3,4]. These mutation rates are expressed per generation and are derived from refs. 3,4 using a generation time of 25 years. They are within the range of values used in the literature ([5.3–10.3] $\times 10^{-10}$ mutations/nucleotide/year for the Y chromosome, and [1.30–2.74] $\times 10^{-8}$ mutations/nucleotide/year for mtDNA, according to SI Table 8.2 in ref. 98, giving a summary of mutation rates used in the literature). As generations do not overlap in the model, males and females have the same generation time, so the same value was used for Y chromosome and mtDNA. Sampling and conversion to VCF are achieved by using tskit v 0.5.6.

### $\pi$-based estimation of effective population size

In each simulation, 20 individuals per village were sampled every 20 generations from the five villages. Effective population sizes of males and females were estimated using Nei's nucleotide diversity ($\pi$)[99], which corresponds to the average pairwise number of differences per site between each pair of sequences within a sample. Y-chromosome and mitochondrial nucleotide diversities were calculated in two ways: by averaging intra-village nucleotide diversities across the five villages ($\pi$), and by calculating nucleotide diversity on pooled samples from all villages (global $\pi$). Then, male and female effective population sizes were obtained by dividing nucleotide diversity with the mutation rate used in the simulations (see Eqs. (4) and (5)). Mean ratios of female-to-male effective population sizes $N_e^{mt/Y}$ (Eq. (6)) and their 95% bias-corrected and accelerated (BCa) confidence intervals were computed by running a bootstrap with 10,000 re-samplings of the set of effective population sizes for all replicates, every 20 generations, using the R package 'boot'[100,101]. $N_e^{mt/Y}$ is expected to be equal to 1 in a panmictic population of constant size.

$$N_e^{mt} = \frac{\pi_{mt}}{2 \times \mu_{mt}} \tag{4}$$

$$N_e^{Y} = \frac{\pi_{Y}}{2 \times \mu_{Y}} \tag{5}$$

$$N_e^{mt/Y} = \frac{N_e^{mt}}{N_e^{Y}} \tag{6}$$

Reduction factors of male effective population size were calculated by dividing the initial number of males in the simulation (i.e. 750) by the estimated male effective population size 100 generations after the introduction of the kinship rule of interest for scenarios 1 to 2h, and

100 generations after the introduction of patrilineality for scenarios 3a to 3c. Maximum female-to-male effective population size ratios were also calculated at the same times.

### Coalescent-based estimation of effective population size

Tree sequences obtained from SLiM simulations were sampled by randomly selecting 100 individuals (10 males and 10 females in each of the five villages) 100 generations after $t_0$, and then converted into Nexus files using the package tskit v 0.5.6[96] and a custom python script. Nexus files were then converted into XML files using BEASTGen v 0.3[59]. BSPs were generated from simulated data using BEAST v 1.10.5[59]. A piecewise constant model with ten groups was selected and the Jukes–Cantor (JC) substitution model was used, as set in the simulations. Clock rates were set according to mutation rates used in the simulations i.e. $2.5 \times 10^{-8}$ mutations/nucleotide/generation for the Y chromosome and of $5.5 \times 10^{-7}$ mutations/nucleotide/generation for mtDNA. These rates have been converted to mutations per nucleotide per year using a generation time of 25 years. Priors for root height and population size were set to a uniform distribution of parameters [0, $10^{10}$]. A MCMC chain of 5,000,000 iterations, with a sampling every 1000 steps was run for each simulation for the Y chromosome and mtDNA. Skyline plots were generated using the script plot-skyline.R available on Art Poon's github[102].

For each scenario, the average male and female effective population sizes over time were computed by averaging the median BSPs obtained using the Y chromosome and mtDNA, respectively, across all replicates. In addition, reduction factors of male effective population size were calculated by averaging the ratios of the initial number of males in the simulation (i.e. 750) to the minimum coalescent-based male effective population size across replicates. The maximum female-to-male effective population size ratio corresponds to the peak of the average BSP of female-to-male effective population size over replicates. We chose this estimator of the maximum female-to-male effective population size ratio so that it is consistent with the graphical representation in Figs. 5, 6 and Supplementary Figs. 12–14. As peaks do not occur at the same time in each replicate, this value is underestimated compared to the average of the peaks across replicates (Supplementary Fig. 15).

### Reporting summary
Further information on research design is available in the Nature Portfolio Reporting Summary linked to this article.

## Data availability
All data were generated by simulation and can be reproduced by running the scripts available on github at: https://github.com/lea-guyon/PatriSim and on Zenodo at: https://doi.org/10.5281/zenodo.10854123[103]. All means and their 95% confidence intervals are presented in Figs. 2–6 and Supplementary Figs. 3 to 17 and 26 to 28 were computed from simulated data and are provided in the Source Data file. Source data are provided with this paper.

## Code availability
Scripts used to run the simulations and analyse the simulated genetic data are available on github at: https://github.com/lea-guyon/PatriSim and on Zenodo at: https://doi.org/10.5281/zenodo.10854123[103].

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

## Acknowledgements

L.G. was supported by a fellowship from the Ecole normale supérieure (ENS Paris): CDSN. J.G. was supported by a French National Centre for Scientific Research (CNRS) fellowship: 80Prime (TransIA). We thank Philippe Chambon, Aline Thomas and Michael Houseman for helpful discussions. Any errors are our own.

## Author contributions

The project was conceived by R.C. and E.H. R.C., E.H. and L.G. designed the socio-demographic model with inputs from J.G. and B.T. L.G. wrote scripts with contributions from J.G. L.G. run all simulations and analysed data. L.G. and R.C. wrote the paper, with inputs from E.H., J.G. and B.T.

## Competing interests

The authors declare no competing interests.
