## [Peer Review File · Nature Communications]

Patrilineal segmentary systems provide a peaceful explanation for the post-Neolithic Y-chromosome bottleneckReviewers' Comments:

Reviewer #1:

Remarks to the Author:

In this manuscript, Guyon et al. explore a hypothesis for the observed reduction in human male effective population sizes dated to several millennia ago. Karmin et al. 2015 hypothesized that increased variance in male reproductive success associated with cultural changes during the Neolithic transition drove this bottleneck, which has been observed in human populations across the globe. Additionally, some have hypothesized that, during this time period, increased violent conflict among patrilineal clan groups could have driven a high rate of extinction of Y-chromosome lineages. Noting that different kinship systems can, on their own, reduce diversity on uniparental lineages, the authors develop a model where they can test the effect of different kinship structures on Y-chromosome vs. mtDNA diversity/effective population size estimates

I think there is value in applying new simulation methods, like SLiM, to model complex sociodemographic scenarios and test their impact on population genetic variation. I appreciate that the authors sought to mechanistically test Karmin et al's hypothesis with thoughtful modelling. It is also good to see the authors refer to the anthropological literature to establish their parameter space.

It seems clear that variance in reproductive success among patrilineal descent groups, especially combined with lineal fission, can produce a significant dip in Y-chr Ne. However, it remains difficult to determine which model best fits the patterns observed in real data from across multiple global contexts. While the authors are able to generate Y-chr bottlenecks without explicitly modelling violence, they report no way of distinguishing between violent and non-violent sources of reproductive variance.

Additionally, I find the overall range of models considered to be limited and unrealistic, with a baseline that assumes that human populations shifted from having no structure at all to some type of patrilocal residence-based system.

I outline my concerns with this paper in more detail below:

Distinguishing between scenarios with and without violence

The authors' results suggest that while a shift to a kinship system based on patrilocal residence can reduce male/Y-chromosome effective population size, this cultural change alone is not sufficient to explain the patterns observed in real human genomic data. They find that both significant variance in reproductive success (for non-violent reasons) and/or increased rates of violent male death result in the most marked declines in Y-chr Ne. Furthermore, when there is variance in reproductive success, a lineal fission pattern can further exacerbate this decline.

Variance and violence are modelled separately here. However, as far as I can tell, both have the same effect on Ne estimates, i.e. they cause a bottleneck. The strength and timing of the bottleneck all vary based on the specific parameter values used, but ultimately it is not really possible to distinguish between models with and without violence by looking at the resulting genomic data. Therefore, the utility of this model for learning about human population history is inherently limited.

While it is true that variance in reproductive success among clans could be caused by many factors other than overt violence, I think that the choice to consider violence as fundamentally distinct from, for instance, differential social status or access to resources, is somewhat arbitrary. Throughout history, human groups have used violence to gain and/or maintain social dominance and/or access to resources. Therefore, I find it hard to imagine scenarios where these factors would not co-occur.

Assumption of universal adoption of patrilocality

The authors offer little independent support for its preferred hypothesis, while minimizing the importance of violence in human history. For example, they state that "it is a strong hypothesis to assume that violence was widespread around the world at those times" (line 26-27) – however, their alternative explanation requires that almost all human societies adopted a particular mode of kinship, all around roughly the same time. It isn't clear to me why this is not an equally "strong hypothesis", as there is currently significant variation in kinship systems among human societies. In the context of a world being shaped by widespread societal, demographic, technological, and climatic changes, I don't think it is too far-fetched to imagine that violent conflict over resources could have become more commonplace.

It is also plausible that, at the same time, many societies shifted to new kinship systems. However, I feel that the authors' proposal of a near-universal shift to, specifically, patrilocal residence systems warrants more examination. Why would this have been the preferred structure for transitioning hunter-gatherers? The authors mention that kinship systems and subsistence strategies tend to covary across human societies, but the possible reasons for this are not discussed. I also suspect there is significant variation among human societies.

I also want to point out here that an assumption of recent panmixia prior to the establishment of patrilocal kinship systems is unreasonable – modern hunter-gatherer societies exhibit a diverse range of robust kinship systems, and there is no reason to believe ancient hunter-gatherers were any different. It is unclear how this assumption impacts the results, but it should be tested more explicitly.

Missing discussion of major demographic events

Long-range mass migrations were a major driver of cultural and demographic change during this period. However, discussions of these kinds of events, which are well characterized in the literature, are lacking in this paper. For example, studies have found a large influx of genetic ancestry from Yamnaya steppe populations into European populations dating to around 4-5kya, appearing to coincide with the observed decline in Y-chr Ne.

It is highly likely that these kinds of migrations, which brought genetically diverged groups into closer proximity, and perhaps into competition over resources and power, would have driven an increase in violent conflict. While the authors focus exclusively on models where the kinship systems are relatively closed and stable, I believe that 'non-equilibrium' events cannot be discounted when trying to explain worldwide Y-chr bottlenecks. These might be modeled through the stochastic introduction of new, diverged 'villages' into the kinship system.

Range of models

Overall, I think the authors must explore a wider range of models to strengthen the validity of their results. They focus on models that are constructed to minimize Y-chr Ne while neglecting plausible scenarios that might counteract the stochastic loss of Y-chr lineages (e.g. low levels of male migration, or fusion of descent groups). Additionally, as previously mentioned, I believe their baseline model of panmixia until 100 generations ago is unlikely to be true. If human populations exhibited structure prior to patrilocality, would the dip in Y-chr Ne be as pronounced?

Additionally, the authors should test models where there has been no shift in kinship system, but where there is increased variance in male reproductive success (by violence or other means). The authors seem to suggest that a change in the way humans structured their societies was necessary for producing a dip in Y-chr Ne, but to what extent are these dips possible without a change in structure? Similarly, I think the authors should model bilateral descent with increased variance in male reproductive success. A model of lineal fission with violence and without other sources of variance in reproductive success seems to have been skipped.

To summarize, I think the authors should model a more complete range of scenarios (with and without increased variance in male reproductive success), and should model structure in human populations going back much longer than 100 generations. This would further validate their main argument that both a shift to patrilineal clan structures AND an increase in variance in male reproductive success are necessary to drive the observed decline in male N_e .

Mutation rates

The authors point out the use of slightly different mutation rates by previous authors, and different estimates of the timing of the Y-chr bottleneck. These rates are on the order of 10^{-9} , but Guyon et al. use a much faster mutation rate – 2.5×10^{-8} – in this work. Can the authors comment on this choice? If a faster rate is more realistic, what are the implications for previous estimates of the timing of the Y-chr bottleneck? What rates have been previously assumed for mtDNA evolution? Is there any variation in those rates that might affect the timing of changes in the mt/Y N_e ratio?

Reviewer #2:

Remarks to the Author:

I find this article very well-thought out and convincing, and it deserves publication. Its strongest point is a thoughtful integration of population genetics, modelling techniques, archaeogenetic data and social context. It is particularly important because it contributes to an area where there has been a widespread tendency by both geneticists and archaeologists to follow superficial and uncritical readings of genetic data and to short-cut important steps such as population-level modelling.

The comments are minor points which will improve the argument and discussion.

- P. 2-3, review of literature documenting high levels of violence in the Neolithic. There is an additional point here which supports the authors' argument but which is not highlighted in the text and should be. This is that the archaeological and skeletal evidence reviewed shows high levels of violence in the Neolithic, which begins 2-3 millennia before the decrease in male-linked genetic diversity; if violence were responsible for the genetic shift, we should see the genetic shift from the beginning of the Neolithic, not from the beginning of the Bronze Age.
- P. 2-3. It is also worth pointing out that skeletal studies have consistently failed to show a substantial increase in violence in the Bronze Age compared to earlier periods. It is now widely acknowledged that the widespread prevalence of weapons as male accoutrements in the Bronze Age has to do with social and political fashions, and may not have anything to do with actual levels of violence. A. Harding has discussed the social persona of the warrior as an emerging figure in the Bronze Age.
- P. 3 lines 65-75. It might also be worth discussing the "stochastic loss of lineages" here; it seems to be implied but not actually mentioned. In any situation tracing lineages of any kind over time from a putative Time 0, some of them will go extinct and others will expand, purely due to random stochastic factors.
- P. 4 lines 118-124. This could be described a bit more clearly.
 - o (i) if you start with 1500 individuals and have 1% growth per generation over 2500 generations, you end up with considerably more than the current population of the earth. So does the population growth start only after the 2500 generations of panmixia? This seems to be implied in the diagram (Figure 1). Even over only 100 generations, 1% growth is still quite high, resulting in more than doubling the population.
 - o (ii) why do you need to model population growth at all, rather than having the population being in a stable demographic regime? Put in some justification of why the model was done this way.
- It would be good to have some discussion of timing and speed. From the figures in this manuscript, the bottleneck in male genetic diversity would become most evident 40-60 generations (perhaps 1000-1500 years) after the change in lineality rules. The main difference between the "with violence"

and “without violence” scenarios is speed: “with violence” scenarios are faster. I understand that Kamin et al. reconstructed both the existence of the bottleneck and the timing of it using modern genetic data, and the timing is entirely based on projected trend rates; it is not calibrated to any actual ancient data points from aDNA and the timing of the transition could be rather approximate. Indeed, in many areas, it corresponds better to the onset of the Neolithic (and perhaps new lineality associated with new property relations) than to the beginning of the Bronze Age. So this could be mentioned, not as a way of distinguishing between the two scenarios using actual currently-existing data, but as a way which could be developed in the future of distinguishing between them by using aDNA to get estimates of genetic diversity at specific dated moments in the past.

- One additional factor that could be mentioned in discussion: the effect of violence here is based on how many males it removes from the reproducing population. It has nothing to do with who is doing the violence or where it comes from. So in that sense, it would be the same whether increased violence came about from an invading horde of warlike people (as in the most extreme of the “Yamnaya invasion” scenarios) or in more realistic and plausible scenarios of endogenous social change without genetic replacement.

- Throughout: it would be good to modify the language so that the manuscript does not say that villages or people “exchanged women”; instead it can say that women “moved between villages”. Talking about “the exchange of women” is anthropologically traditional but it implies that women are passive objects and men or groups as a whole are the active agents in society.

Please note: as part of this review, I protest Springer Nature's exclusionary policies of high pricing and restrictive open access for a product which is created through donated labour from authors, reviewers and editors; I am donating my labour in this review out of support for colleagues, including the authors and readers, and fairness demands that this should be passed on throughout the system.

Reviewer #3:

Remarks to the Author:

This paper offers an interesting alternative to the model of Zeng et al. (2018), which was developed in an attempt to explain the data of Karmin et al. (2015) showing a seventeen to one ratio of females to males in post-Neolithic European humans. Zeng et al. incorporated cultural transmission belonging to Y chromosome haplogroups that competed with one another ecologically, resulting in the demise of enough Y-chromosome haplotypes to produce an extremely female-biased population.

Guyon et al. develop a numerical simulation of Y chromosomes and Mitochondria that involve an array of demographic assumptions.

1. The overall population size in the model simulation is 1,500, which is divided into five “villages” of 300 each.
2. In one model there is random reproduction between males and females among the 300. It is not explained how the sex ratio at birth is determined; it is not stated that there are 150 males and 150 females at the start of the simulation in each group.
3. In the second model, each group of 300 is divided into three groups of 100. It is not stated whether there are initially fifty males and fifty females in each descent group.
4. The growth rates of descent groups are chosen from a normal distribution initially. It is unclear what happens when (and if) a descent group has ninety-nine males and one female, for example. It is stated that descent groups “undergo fusion” when their size exceeds 100. But they start at 100? If a descent group goes extinct, what happens to the system? Can all descent groups go extinct? Do males and females have the same birth rates?
5. Female migration. If the female migration rate between villages is 10% (line 400), does this mean on average fifteen females from each village move to a different village, and are these females distributed equally among the five descent groups?
6. There have to be detailed assumptions on reproduction between males and local females and

between males and immigrant females. What does minimizing "the number of individuals to remain single" entail? "Descent groups display different growth rates." There are fifteen descent groups. Once an r_i is chosen, how are the mating pairs chosen?

7. How many descent groups are eventually seen? If the descent groups get too big, can the sex ratio in a split-off group become extreme?

All of the above issues pertain only to the generation-to-generation samplings, not to the estimates of $N_e^{(Y)}, N_e^{(mt)}$ obtained from n and BSP. These also involve assumptions on the mutation rates for Y and mt , in n and BSP. Assuming the same numbers were used, there is an enormous discrepancy in the male N_e and the female-to-male effective sizes using the two methods. This difference is not discussed. In addition, the value "17" from Karmin et al. does not appear in Table 2, and in Figure 5 the average is 8.86. How does this compare with Zeng et al.? The authors call Zeng et al.'s model "Violence" but do not say how that model is included in the simulation to produce Figure 4. Are the cultural groups of Zeng et al. equivalent to the descent groups here?

I think the paper loses its impact by not having a table of parameters and their values. It is too important to be left to a supplement. I believe more care in the writing could produce an acceptable manuscript. It is a very interesting topic.

Minor points

Line 21. Our paper. Is this the present manuscript? It refers to Fox which is (10).

Lines 50-51. "This was reported" — not "Such observation".

Line 57 and elsewhere. "affiliated with" not "to"

Lines 73, 81. "This variance" — not "Such variance"

Line 88. Define "segmentary patrilineal system"

Lines 118-124. The mating system is not clear. How are mates chosen?

What happens if the sex-ratio is not even?

The graphs go back 125 generations. Why?

Line 134 and elsewhere. Use "exceeds" everywhere instead of "overpasses"

Line 192. "dropped by a factor of 27.6.."

Figure 5, legend. "...Y chromosome and mtDNA effective population sizes under.."

We would like to thank all the reviewers for their highly relevant comments, which have helped us to significantly improve our model and paper. All changes appear in blue in the manuscript.

REVIEWER COMMENTS

Reviewer #1 (Remarks to the Author):

In this manuscript, Guyon et al. explore a hypothesis for the observed reduction in human male effective population sizes dated to several millennia ago. Karmin et al. 2015 hypothesized that increased variance in male reproductive success associated with cultural changes during the Neolithic transition drove this bottleneck, which has been observed in human populations across the globe. Additionally, some have hypothesized that, during this time period, increased violent conflict among patrilineal clan groups could have driven a high rate of extinction of Y-chromosome lineages. Noting that different kinship systems can, on their own, reduce diversity on uniparental lineages, the authors develop a model where they can test the effect of different kinship structures on Y-chromosome vs. mtDNA diversity/effective population size estimates

I think there is value in applying new simulation methods, like SLiM, to model complex sociodemographic scenarios and test their impact on population genetic variation. I appreciate that the authors sought to mechanistically test Karmin et al's hypothesis with thoughtful modelling. It is also good to see the authors refer to the anthropological literature to establish their parameter space.

It seems clear that variance in reproductive success among patrilineal descent groups, especially combined with lineal fission, can produce a significant dip in Y-chr N_e . However, it remains difficult to determine which model best fits the patterns observed in real data from across multiple global contexts. While the authors are able to generate Y-chr bottlenecks without explicitly modelling violence, they report no way of distinguishing between violent and non-violent sources of reproductive variance.

It is true that our model does not allow us to distinguish between violent and non-violent scenarios to explain the Y-chromosome bottleneck. However, it does show that violence resulting in the killing of 15% of males per generation is neither necessary nor sufficient to explain this bottleneck, and that non-violent sources of variance in reproductive success are required to explain the strength of the male effective size bottleneck. Furthermore, our main goal here is not to decide whether or not there was violent intergroup competition but to broaden the possible explanations for the bottleneck by proposing alternative plausible scenarios. As discussed below, scenarios with violence reduced male effective size faster than scenarios without violence, which may help us to distinguish between the two in the future.

Additionally, I find the overall range of models considered to be limited and unrealistic, with a baseline that assumes that human populations shifted from having no structure at all to some type of patrilocal residence-based system.

We thank the reviewer for this highly valuable comment, which has greatly helped to improve the article. As explained in more detail below, we have implemented a new scenario with a first transition to a bilateral system and a second transition to a patrilineal system. We still observe a significant reduction in male effective size after the introduction of patrilineal descent, and female-to-male effective size ratios consistent with the value of 17 reported by Karmin *et al.*¹.

I outline my concerns with this paper in more detail below:

Distinguishing between scenarios with and without violence

The authors' results suggest that while a shift to a kinship system based on patrilocal residence can reduce male/Y-chromosome effective population size, this cultural change alone is not sufficient to explain the patterns observed in real human genomic data. They find that both significant variance in reproductive success (for non-violent reasons) and/or increased rates of violent male death result in the most marked declines in Y-chr N_e . Furthermore, when there is variance in reproductive success, a lineal fission pattern can further exacerbate this decline.

Variance and violence are modelled separately here. However, as far as I can tell, both have the same effect on N_e estimates, i.e. they cause a bottleneck. The strength and timing of the bottleneck all vary based on the specific parameter values used, but ultimately it is not really possible to distinguish between models with and without violence by looking at the resulting genomic data. Therefore, the utility of this model for learning about human population history is inherently limited.

Indeed, modern genetic data alone may not be sufficient to decide between the different scenarios. Our main point in this paper is that violent conflict resulting in the killing of large numbers of males is neither necessary nor sufficient to achieve the bottleneck reported by Karmin *et al.*¹. So far, only one scenario involving violent competition between human groups has been proposed to explain the bottleneck in male effective size. It seems important to us to broaden this vision by proposing alternative scenarios that may also explain this phenomenon. A combination of genetic, anthropological and archaeological records, considered together, may then certainly help to decide which scenario is most likely to have occurred. In particular, as suggested by reviewer #2, palaeogenetics may help to distinguish between violent and non-violent causes of male effective size decline by computing the genetic diversity at multiple time points in the same population. A paragraph has been added to the discussion (l. 601-607) to address this concern:

"Computations of π -based effective size indicate that violence impacts genetic diversity in a faster way than the kinship system. This outcome implies that future research conducting ancient DNA diversity measurements at multiple time points within the same archaeological site may provide insights into the pace of male effective size reduction, allowing to distinguish between more violent or more peaceful scenarios at local scales, and lead to a more comprehensive picture of the socio-demographic processes at work in ancient human populations."

While it is true that variance in reproductive success among clans could be caused by many factors other than overt violence, I think that the choice to consider violence as

fundamentally distinct from, for instance, differential social status or access to resources, is somewhat arbitrary. Throughout history, human groups have used violence to gain and/or maintain social dominance and/or access to resources. Therefore, I find it hard to imagine scenarios where these factors would not co-occur.

Although the reviewer is right to say that it is generally accepted that competition for resources increases the rate of violence², other factors may also play a role, such as climate change³⁻⁵, and cultural norms⁶. It is therefore difficult to predict rates of violence in our past. Using ethnographic and/or archaeological data, some authors have claimed that Pleistocene hunter-gatherers had low levels of warfare^{2,7}, while others came to the opposite conclusion^{4,8,9}, fuelling the long-standing debate between Rousseauians and Hobbesians about human nature⁶. Furthermore, a recent study based on data from various archaeological sites in the Old World found that levels of violence were stable from the time *Homo sapiens* diverged from other primates until the Bronze Age, but increased significantly during the Iron and the Middle Ages¹⁰. On the other hand, other studies have reported an increase in violence in more ancient times, such as during the Early Neolithic for Central Europe¹¹, and during the Copper Age and Late Bronze Age for the Middle East¹². These inconsistencies likely result from the nature of the archaeological record, which suffers from numerous biases, including the particularly small sample sizes for the earliest periods. Moreover, although a large number of individuals have been found buried with a weapon in the Bronze Age archaeological record, this does not coincide with an increase in interpersonal violence in either Europe or the Middle East, suggesting that the “idealisation” of the “warrior figure” may not have reflected an intensification of warfare¹²⁻¹⁴. In this context of lack of consensus on past rates of violence in our species, our aim is to propose scenarios to explain the Y-chromosome bottleneck that do not require a high rate of violence. Zeng *et al.*¹⁵ hypothesised that 15% of males were killed at each generation, which is among the highest rates of violence recorded in human history¹⁰. Nevertheless, this rate of violence had a limited effect on the male effective size (scenario 2b, coalescent-based female-to-male effective size below 3.6, whether or not male and group migration are included in the model). Non-violent sources of variance combined with lineal fission had a greater effect on male effective size (scenario 2g, coalescent-based female-to-male effective size ratio between 7 and 20). This shows that violence is neither sufficient nor necessary to explain this bottleneck. Of course, we agree that violence, in combination with lineal fission and/or non-violent sources of variance, could have contributed to accelerating the loss of male effective size. However, we believe this is important to broaden the range of scenarios explaining the Y-chromosome bottleneck to show that such a bottleneck is not per se evidence of high levels of violence.

We have clarified this point by adding lines 25-42:

*“However, it is questionable whether violence is necessary to explain the genetic signal observed by Karmin et al. [1] as the level of violence worldwide at these times remains uncertain (Zeng’s model assumes that 15% of males were killed at each generation [9]). Using ethnographic and/or archaeological data, some authors have claimed that Pleistocene hunter-gatherers exhibited low levels of warfare [11, 12], while others came to the opposite conclusion [13–15], fueling the long-standing debate between Rousseauians and Hobbesians about human nature [16]. Furthermore, a recent study based on data from various archaeological sites in the Old World found that levels of violence remained stable from the time *Homo sapiens* diverged from other primates until the Bronze Age, but significantly increased during the Iron and the Middle Ages [17]. On the other hand, other*

studies have reported an increase in violence in more ancient periods, such as during the Early Neolithic in Central Europe [18], and during the Copper Age and Late Bronze Age in the Middle East [19]. These inconsistencies likely result from the nature of the archaeological record, which suffers from numerous biases, including the particularly small sample sizes for the earliest periods. Moreover, although a large number of individuals have been found buried with weapons in the archaeological records of the Bronze Age, this does not coincide with an increase in interpersonal violence in either Europe or the Middle East, suggesting that the idealisation of the social persona of the warrior may not reflect an intensification of warfare [19–21].”

Assumption of universal adoption of patrilocality

The authors offer little independent support for its preferred hypothesis, while minimizing the importance of violence in human history. For example, they state that “it is a strong hypothesis to assume that violence was widespread around the world at those times” (line 26-27) – however, their alternative explanation requires that almost all human societies adopted a particular mode of kinship, all around roughly the same time. It isn’t clear to me why this is not an equally “strong hypothesis”, as there is currently significant variation in kinship systems among human societies. In the context of a world being shaped by widespread societal, demographic, technological, and climatic changes, I don’t think it is too far-fetched to imagine that violent conflict over resources could have become more commonplace.

In this article, our objective is not to minimize the importance of violence but we aim at broadening the range of possible explanations for this bottleneck (see our answer below).

It is also plausible that, at the same time, many societies shifted to new kinship systems. However, I feel that the authors’ proposal of a near-universal shift to, specifically, patrilocal residence systems warrants more examination. Why would this have been the preferred structure for transitioning hunter-gatherers? The authors mention that kinship systems and subsistence strategies tend to co-vary across human societies, but the possible reasons for this are not discussed. I also suspect there is significant variation among human societies.

The Neolithic transition corresponds to a massive change in subsistence systems, with a transition from extractive subsistence systems (hunting and foraging) to productive subsistence systems (pastoralism and agriculture). The relationships between subsistence systems and kinship systems have been reported by many anthropologists. Marlowe² showed that 60% of contemporary non-forager populations are patrilocal, compared with only 34% of forager populations. In addition, 47% of non-foragers, but only 14% of foragers, have patrilineal descent. This has been famously summarized by Aberle¹⁶: “the cow is the enemy of matriliney”. More recently, Holden and Mace confirmed that such relationships between subsistence and kinship systems exist in the Bantu populations even after correcting for the historical relationships between populations¹⁷. Thus, although the reviewer is right in saying that there is variation, an increase in frequency of patrilocality and patrilineality across the globe after the Neolithic transition seems plausible.

Two main hypotheses have been proposed to explain the relationship between subsistence strategies and kinship systems. According to structural-functionalists, patrilocality and patrilineality emerge when resources can be accumulated. Indeed, with movable property, prosperous men can offer a bride price to the parents of marriageable young women, rather

than move in with their in-laws for bride service. In this way, daughters are separated from their parents, while men remain in their place of birth after marriage. movable property is thought to empower men to resist matrilineal and matrilocal traditions^{18–20}. According to evolutionary anthropologists, patriliney increases in frequency in wealth-accumulating societies because such inheritance rule is better suited to maximising reproductive success due to the greater reproductive potential of males compared to females^{17,21}.

We have developed further these points in discussion lines 494-510:

“Two main hypotheses have been advanced to explain the relationship between subsistence strategies and kinship systems. According to structural-functionalists, patrilocal and patrilineality arise when resources can be accumulated. Indeed, with movable property, prosperous men can offer a bride price to the parents of marriageable young women, rather than moving in with their in-laws for bride service. In this way, daughters are separated from their parents, while men remain in their place of birth after marriage. More generally, movable property is thought to empower men to resist matrilineal and matrilocal traditions [59–61]. According to evolutionary anthropologists, patriliney is more prevalent in wealth-accumulating societies because this inheritance rule is better suited to maximising reproductive success due to the higher reproductive potential of males compared to females [62, 63]. In addition, Marlowe [12] showed that patrilocal is over-represented in contemporary non-forager populations (mainly farmers and herders) compared to forager populations (60% and 34% of populations respectively). Similarly, patrilineal descent is over-represented in non-foragers compared to foragers (47 and 14% respectively). This has been famously summarized by Aberle: “the cow is the enemy of matriliney” [41]. More recently, Holden and Mace confirmed that such relationships between subsistence and kinship systems exist in the Bantu populations even after correcting for the historical relationships between populations [63].“

Consistent with this, ancient DNA studies show that reported male relatedness within archaeological sites is often higher than female relatedness. In particular, males share the same haplogroup on the Y chromosome, whereas there are different haplogroups for mtDNA^{22–30}. In addition, reconstructed pedigrees show more first and second-degree related males than females^{22,24,26,28,29,31}. These observations are interpreted as the result of patrilocal residence, with males remaining in their birthplace and females migrating between sites. In addition, several sites showed evidence of paternal transmission of goods and/or social status^{23,24,27,28}, consistent with a patrilocal and possibly patrilineal inheritance system. This idea was included lines 521-531:

“Consistent with the hypothesis of a rapid spread of patrilocal after the Neolithic transition, ancient DNA studies show that male relatedness is often higher than female relatedness within archaeological sites from the Neolithic and the Bronze Age (mostly from Europe). In particular, males tend to share more often the same haplogroup of the Y chromosome, whereas there is a greater diversity of haplogroups for mtDNA [66–75]. In addition, reconstructed pedigrees show more first and second-degree related males than females [66, 68, 70, 73–75]. These observations have been interpreted as the result of patrilocal residence (with varying degrees of compliance with the rule), with males more likely to remain in their birthplace and females more likely to migrate between sites. Furthermore, several sites showed evidence of paternal transmission of goods and/or social status (for example [67, 68, 72, 74]), suggesting patrilineal inheritance systems.”

I also want to point out here that an assumption of recent panmixia prior to the establishment of patrilocal kinship systems is unreasonable – modern hunter-gatherer societies exhibit a diverse range of robust kinship systems, and there is no reason to believe

ancient hunter-gatherers were any different. It is unclear how this assumption impacts the results, but it should be tested more explicitly.

To address this concern, we ran additional simulations, including a first transition from panmixia to a bilateral descent system (which is the descent system the most frequently observed in contemporary hunter-gatherers²) followed by a transition to patrilineal descent. The post-marital residence patterns of late Pleistocene hunter-gatherers are poorly known. It has been argued that they may have had patrilocal residence³², matrilineal residence³³, or multilocal residence². Consequently, we simulated an initial shift from panmixia to a bilateral system with either patrilocal residence, matrilineal residence, or multilocal residence, followed by a shift to a patrilineal system. A diagram of this extended model is shown in Supplementary Figure 2.

These new simulations show that the bottleneck signal due to the patrilineal system is not affected by this initial shift to a bilateral system, whatever the residence rule considered. Furthermore, since population growth starts before the introduction of patrilineality, the female effective size has time to increase before the bottleneck in male effective size occurs, leading to an increase in the female-to-male effective size ratio, which exceeds the reported value of 17 by Karmin *et al.*¹. The results are shown in Figures 4 and 6 and have been integrated in Table 2. Note that values in Table 2 have slightly changed compared to the previous version since we changed the value of the fission threshold (from 100 to 150 individuals, see the answers to reviewer #3). It also changed because we slightly modified the way we estimate the reduction factors of male effective size (see legend of Table 2) so that π -based and coalescent based reduction factors are now more comparable to each other (the reference value is now the same for both methods, i.e. the number of males at t_0).

These results are now discussed in “Discussion” (1.442-458):

*“As the transition to patrilocality and patrilineality is unlikely to have occurred directly from a panmictic population, we explored the evolution of genetic diversity under scenarios involving an initial shift from panmixia to a bilateral system (which is the descent system the most frequently observed in contemporary hunter-gatherers [12]), followed, 100 generations later, by a transition to a patrilineal system. The post-marital residence patterns in pre-Neolithic populations are poorly known. It has been argued that they may have had patrilocal residence [58], matrilineal residence [53], or multilocal residence [12]. Consequently, we simulated an initial shift from panmixia to a bilateral system with either patrilocal residence, matrilineal residence, or multilocal residence, followed by a shift to a patrilineal system (Supplementary Fig. 2). All three scenarios showed the same reduction in male effective size after the introduction of patrilineality, with π -based reduction factors above 19, and coalescent-based reduction factors around 2.8. Therefore, a reduction in male effective size due to patrilineality is compatible with any prior kinship system. These scenarios also showed that when population growth is older than the onset of patrilineality, the bottleneck of male effective size occurs at a time when female effective size is high, leading to coalescent-based female-to-male effective size ratios comprised between 18.00 (matrilineal residence) and 19.89 (patrilocal residence), similar to the value of 17 reported by Karmin *et al* [1].”*

Missing discussion of major demographic events

Long-range mass migrations were a major driver of cultural and demographic change during this period. However, discussions of these kinds of events, which are well characterized in the literature, are lacking in this paper. For example, studies have found a large influx of

genetic ancestry from Yamnaya steppe populations into European populations dating to around 4-5kya, appearing to coincide with the observed decline in Y-chr Ne.

Long-range mass migrations certainly played an important role in shaping the diversity of ancient human populations. They have left distinct imprints on contemporary human genomes. Notably, the frequencies of the Y chromosome haplogroups underwent substantial shifts during the Steppe migration, resulting in the overrepresentation of haplogroup R1b1 in Europeans^{34–36}. Our approach does not formally model these long-range migrations. However, our simulations show that when patrilineal descent is combined with intervillage group migration, the loss in coalescent-based male effective size is greater than when group migration is not possible. Indeed, with group migration, a successful Y-chromosome haplogroup can invade other villages, reducing local Y-chromosome diversity.

A paragraph has been added to the Discussion to address this concern (l. 578-590):

“Long-range migrations, such as the movement of Steppe populations from Eastern Europe to Western Europe and Central Asia, occurred around 5,000 BP, and have left distinct imprints on contemporary human genomes. Notably, the frequencies of the Y chromosome haplogroups underwent substantial shifts during the Steppe migration, resulting in the overrepresentation of haplogroup R1b1 in Europeans [4, 88, 89]. Our approach does not formally model these longrange migrations. However, our simulations show that when patrilineal descent is combined with post-fission intervillage group migration, the loss in coalescent-based male effective size is greater than when this group migration does not occur. Indeed, with group migration, a successful Y-chromosome haplogroup can invade other villages, reducing local Y-chromosome diversity. In other words, if Steppe populations were organised into segmentary patrilineal groups as they migrated, this could explain the rapid spread of a particular Y-chromosome haplogroup and the loss of other Y-chromosome haplogroups. Future research should better characterise these interactions between population demography and kinship systems.”

It is highly likely that these kinds of migrations, which brought genetically diverged groups into closer proximity, and perhaps into competition over resources and power, would have driven an increase in violent conflict. While the authors focus exclusively on models where the kinship systems are relatively closed and stable, I believe that ‘non-equilibrium’ events cannot be discounted when trying to explain worldwide Y-chr bottlenecks. These might be modeled through the stochastic introduction of new, diverged ‘villages’ into the kinship system.

Although these scenarios seem interesting to test, they are beyond the scope of this article. We believe they would merit a separate study on their own, as the effect of many new parameters would have to be characterised.

Range of models

Overall, I think the authors must explore a wider range of models to strengthen the validity of their results. They focus on models that are constructed to minimize Y-chr Ne while neglecting plausible scenarios that might counteract the stochastic loss of Y-chr lineages (e.g. low levels of male migration, or fusion of descent groups). Additionally, as previously mentioned, I believe their baseline model of panmixia until 100 generations ago is unlikely to

be true. If human populations exhibited structure prior to patrilocality, would the dip in Y-chr Ne be as pronounced?

We thank the reviewer for this very valuable comment. We now consider the effect of two additional processes: a small proportion (2% per generation) of intervillage male migration (which mimics the effect of adoption, adultery and flexibility in the residence rule), and the possibility that, after a fission, the smallest of the two newly born descent groups migrate to another village. Both of these processes have been described in the literature^{37–40}.

In the case where 2% of males migrate to another village in each generation, our simulations show that, as predicted by the reviewer, the Y-chromosome bottleneck was less severe than in the absence of such migration. However, even considering such migration, the effect of a non-violent patrilineal system (with variance in reproductive success and lineal fission) is still stronger than the effect of violence (6.07 versus 1.79 for the coalescent-based female to male effective sizes ratio, as shown in Supplementary Table 4).

On the other hand, when post-fission group migration is taken into account, the coalescent-based male effective size is reduced even further than without such migration, and the female-to-male effective size ratio reaches 21.44 in a non-violent patrilineal system, with variance in reproductive success and lineal fission (Supplementary Table 5). This may seem counterintuitive, but this is because such migration allows successful patrilineal descent groups (together with their Y haplogroup) to migrate to other villages, replacing local Y haplogroups in the long term, thereby reinforcing the Y chromosome bottleneck.

see lines 426-441:

“Scenarios considering 2% intervillage male migration and/or post-fission migration showed similar relative results compared to scenarios without these types of migration, with an important effect of variance in reproductive success between groups and lineal fission on the reduction in male effective size. Adding 2% of male migration (mimicking adoption, adultery and flexibility in the residence rule) mitigated the reduction in both π -based and coalescent-based male effective size (Supplementary Table 4, Supplementary Fig. 3, 12). Adding post-fission migration to the model, mitigated the reduction in the π -based male effective size (Supplementary Table 5, Supplementary Fig. 4). On the other hand, it reduced the coalescent-based male effective size more severely than scenarios without post-fission migration, reaching a reduction factor of 7.87 in scenario 2h (Supplementary Table 5, Supplementary Fig. 13). This means that the migration of some descent groups to the other villages accelerates the reduction of coalescent-based male effective size. Indeed, migrating descent groups experiencing demographic growth may spread their Y-chromosome haplotype throughout the whole population, reducing the overall coalescent-based male effective size. However, because it introduces new Y-chromosome haplotypes into villages, this post-fission migration reduces the π -based male effective size less than when the post-fission migration is not included in the model.”

Additionally, the authors should test models where there has been no shift in kinship system, but where there is increased variance in male reproductive success (by violence or other means). The authors seem to suggest that a change in the way humans structured their societies was necessary for producing a dip in Y-chr Ne, but to what extent are these dips possible without a change in structure? Similarly, I think the authors should model bilateral descent with increased variance in male reproductive success. A model of lineal

fission with violence and without other sources of variance in reproductive success seems to have been skipped.

In a monogamous system, increased variance in male reproductive success would also increase the variance in female reproductive success and would not result in different patterns on the Y chromosome and mtDNA. Therefore, increased variance in male reproductive success (but not in female reproductive success) in the absence of patrilineal structure would be the result of a polygynous system where males can mate with multiple females. To assess the effect of polygyny on male and female effective sizes, we modeled a highly polygynous patrilocal population (see Supp fig 21) based on the Kipsigis of Kenya, in which men have any number of wives as long as they can pay bride wealth⁴¹. Some men have up to 12 wives^{42,43} and on average 83% of women have co-wives⁴⁴. The decrease in male effective size assessed by pairwise nucleotide differences within villages was weaker under this Kipsigis-like polygyny compared to a patrilineal system with lineal fission and variance in reproductive success between groups (3.96 and 21.10 respectively). Similarly, the coalescent-based male effective size decreased less in this bilateral and polygynous system than in the patrilineal system (1.59 and 2.77 respectively), resulting in a lower female-to-male effective sizes ratio (see Supplementary Fig 27, 28). Therefore, polygyny alone cannot explain the post-Neolithic bottleneck.

In addition, kinship analyses based on ancient DNA data from the European and Bronze Age Neolithic showed little evidence for half-brothers and half-sisters^{22–25,27,28,31}, which suggests that polygyny was not widespread in those times in Europe.

These observations are commented on in “Discussion” (1.540-560):

“Although we have not included them in our model, other socio-cultural processes may have contributed to the post-Neolithic bottleneck, such as polygyny, which exists in more than 80% of present day human populations [77]. To assess the effect of high levels of polygyny on the reduction in male effective size, we implemented an additional scenario by adding polygyny in a bilateral and patrilocal population. Polygyny was modelled following the case of the Kipsigis of Kenya mating system (Supplementary Table 11). The Kipsigis are one of the most polygynous societies in the world [78]. Men have any number of wives as long as they can afford to pay the bride price [79]. Some men have up to 12 wives [80, 81] and on average 83% of women have co-wives [82]. In our “Kipsigis-like” model, males can marry multiple females, where the number of wives per male follows a geometric distribution with a parameter $p = 1/2$. Male effective size under the “Kipsigis-like” scenario, did not decrease as much as in patrilineal populations undergoing lineal fissions with violence and/or variance in reproductive success between descent groups. The π -based and coalescent-based male effective size reduction factors reached 3.96 and 1.59 respectively (Supplementary Fig. 27, 28). This result is consistent with previous studies indicating that polygyny alone is expected to have a reduced effect on genetic diversity [22, 83], and would probably not be sufficient on its own to account for the male effective size bottleneck reported by Karmin et al. [1]. In addition, kinship analyses based on ancient DNA data from the European and Bronze Age Neolithic showed little evidence for half-brothers and half-sisters [66-69, 72–74], with a few exceptions [70, 75] that have been interpreted as evidence for polygyny or serial monogamy, which suggests that polygyny was not widespread in those times in Europe.”

Furthermore, the scenario of patrilineal descent with lineal fission and violence but no other sources of variance has been added and is commented on in the “Results” section (scenario 2d). This scenario showed a stronger decrease in male effective size than all scenarios with

random fission (without variance: 2a to 2c, and with variance: 2e and 2f), but a weaker decrease than scenarios 2g and 2h (with variance in reproductive success between groups and lineal fission). These results are consistent with a stronger effect of variance in reproductive success between groups on the reduction of male effective size compared to violence.

To summarize, I think the authors should model a more complete range of scenarios (with and without increased variance in male reproductive success), and should model structure in human populations going back much longer than 100 generations. This would further validate their main argument that both a shift to patrilineal clan structures AND an increase in variance in male reproductive success are necessary to drive the observed decline in male N_e .

We believe that our additional scenarios allow us to conclude that increased variance in male reproductive success and lineal fission are necessary to explain the bottleneck, and that this pattern is robust to a wide range of kinship systems preceding patrilineality. In addition, adding flexibility to the residence rule mitigated the reduction in effective male size, but does not change our conclusions. On the contrary, introducing post-fission migration of descent groups into our model strengthened our conclusion that bottlenecks compatible with Karmin *et al.*¹ are achievable under scenarios without violent intergroup competition.

Mutation rates

The authors point out the use of slightly different mutation rates by previous authors, and different estimates of the timing of the Y-chr bottleneck. These rates are on the order of 10⁻⁹, but Guyon *et al.* use a much faster mutation rate – 2.5 x 10⁻⁸ – in this work. Can the authors comment on this choice? If a faster rate is more realistic, what are the implications for previous estimates of the timing of the Y-chr bottleneck? What rates have been previously assumed for mtDNA evolution? Is there any variation in those rates that might affect the timing of changes in the mt/Y N_e ratio?

This discrepancy is due to the fact that mutation rates are expressed in different units. The mutation rates from previous publications are expressed in mutations/bp/year, whereas the mutation rates in this study are expressed in mutations/bp/generation. It is possible to switch from one to the other using a generation time of 25 years. Here we used the mutation rates given by Batini *et al.*^{34,45} ([5.3 - 10.3] x 10⁻¹⁰ mutations/bp/year for the Y chromosome, and [1.30 - 2.74] x 10⁻⁸ mutations/bp/year for mtDNA) which are within the range of values used in the literature (summarised in the SI Table 8.2 in Skov *et al.*⁴⁶).

We have added a few lines in “Material and Methods” to avoid any confusion (l.721-728):

*“Mutations are generated along the coalescent trees with the Python package msprime v 1.1.0 [95] using a mutation rate of 2.5 × 10⁻⁸ mutations/nucleotide/generation for the Y chromosome and of 5.5 × 10⁻⁷ mutations/nucleotide/generation for mtDNA [3, 4]. These mutation rates are expressed per generation and are derived from Batini *et al.* [3, 4] using a generation time of 25 years. They are within the range of values used in the literature ([5.3–10.3]×10⁻¹⁰ mutations/nucleotide/year for the Y chromosome, and [1.30–2.74]×10⁻⁸ mutations/nucleotide/year for mtDNA, according to SI Table 8.2 in Skov *et al.* [96], giving a summary of mutation rates used in the literature).”*

In addition, the mutation rate chosen in our simulations should not affect our estimates of effective sizes and timing of bottleneck since:

- π -based male and female N_e are corrected by dividing the obtained value of estimated nucleotide diversity (π) by the mutation rate, and the mutation rate used to correct the nucleotide diversity is the same as the mutation rate used in the simulations.
- BEAST software takes into account the mutation rates of the Y chromosome and mtDNA when inferring male and female effective sizes over time, and the mutation rates entered in the BEAST software are the same as the mutation rates used in the simulations.

Therefore, whatever value of the mutation rate we choose in the simulations, it will not affect the effective sizes nor the timing of the bottleneck that we infer because we use the same mutation rate.

However, when using real population data, it is possible that the mutation rate chosen in BEAST inference or to correct for nucleotide diversity is different from the real mutation rate. Choosing a faster rate would lead to a more recent bottleneck than the real bottleneck. For example, Batini *et al.*⁴⁵ used a higher mutation rate than Karmin *et al.*¹ and inferred a more recent bottleneck in male effective size. See Introduction (1.9-14):

“The estimated timeframe of this bottleneck varies between world regions, ranging from 8,300 BP in the Near East to 1,400 BP in Siberia, and was estimated to 5,000 BP in Europe according to Karmin et al. [1] using a mutation rate of 0.74×10^{-9} mutations/bp/year for the Y chromosome. Using a slightly higher mutation rate (1.0×10^{-9} mutations/bp/year), Batini et al. [3] estimated more recent dates for this bottleneck (ranging from 4,200 to 2,100 BP for the European and Near Eastern populations) [3].”

Reviewer #2 (Remarks to the Author):

I find this article very well-thought out and convincing, and it deserves publication. Its strongest point is a thoughtful integration of population genetics, modelling techniques, archaeogenetic data and social context. It is particularly important because it contributes to an area where there has been a widespread tendency by both geneticists and archaeologists to follow superficial and uncritical readings of genetic data and to short-cut important steps such as population-level modelling.

The comments are minor points which will improve the argument and discussion.

- P. 2-3, review of literature documenting high levels of violence in the Neolithic. There is an additional point here which supports the authors' argument but which is not highlighted in the text and should be. This is that the archaeological and skeletal evidence reviewed shows high levels of violence in the Neolithic, which begins 2-3 millennia before the decrease in male-linked genetic diversity; if violence were responsible for the genetic shift, we should see the genetic shift from the beginning of the Neolithic, not from the beginning of the Bronze Age.

This is a good point, however the Skyline plots (BEAST) obtained from simulated data showed that there is a latency period of about 70 generations between the introduction of the kinship system of interest (with or without violent intergroup competition) and the

onset of the bottleneck. This observation suggests that the change in kinship system or the increase in violence behaviors may have happened long before the bottleneck detected with BEAST by Karmin *et al.*¹ We clarify this point in lines 459-489:

*“In Karmin *et al.*’s study [1], the male effective size bottleneck was observed after the Neolithic revolution, and was dated at 8,300 years BP in the Near East, 5,000 years BP in Europe and 1,400 years BP in Siberia, which corresponds to 332, 200 and 56 generations ago respectively using a generation time of 25 years. Using a slightly higher mutation rate, Batini *et al.* [3] dated this bottleneck at 4,200 and 2,100 BP (i.e. 140 and 70 generations ago) for the European and Near Eastern populations respectively [3]. The Bayesian skyline plots produced from simulated data under the scenario 2g (patrilineal descent, lineal fission, variance in reproductive success, no violence) show that the male effective size bottleneck reached its maximum around 30 generations before present, 70 generations after the introduction of the patrilineal and patrilineal kinship rules in the population (Figure 5c). In order to understand which scenario(s) could result in a timing for the bottleneck that is compatible with the timing reported by Karmin *et al.* [1] and Batini *et al.* [3], we simulated three additional scenarios. The first scenario (4a) is an extension of the scenario 2g, where patrilineal descent is introduced 200 generations ago (instead of 100). The second scenario (4b) models a first transition from panmixia to a patrilineal and patrilineal system (with the same settings as in scenario 2g), 200 generations ago, followed by a transition to a bilateral and patrilineal system, 100 generations ago. The third scenario (4c) models a first transition from panmixia to a patrilineal and patrilineal system (with the same settings as in scenario 2g), 200 generations ago, followed by a transition to a patrilineal and patrilineal system without variance in reproductive success between groups (as in scenario 2c), 100 generations ago. The Bayesian skyline plots obtained using BEAST are presented in Supplementary Fig. 26. Scenario 4a produced a Y chromosome bottleneck that reached its minimum 40 generations ago (i.e. around 1,000 years ago using a generation time of 25 years), while the scenarios 4b and 4c produced a bottleneck around 130 and 110 generations ago (i.e. 3,250 and 2,750 years ago) respectively, which is more consistent with the timing of bottlenecks inferred from real data by Batini *et al.* [3] and Karmin *et al.* [1]. This suggests that for the bottleneck to have occurred around 3,000-5,000 years ago, an initial transition to kinship systems that are efficient at reducing Y-chromosome (such as the patrilineal scenario 2g, with lineal fission and variance in reproductive success between groups) must have been followed by a transition to kinship systems that are less efficient at reducing Y-chromosome diversity (here we considered a bilateral descent system or a patrilineal system with no variance in reproductive success between groups).”*

- P. 2-3. It is also worth pointing out that skeletal studies have consistently failed to show a substantial increase in violence in the Bronze Age compared to earlier periods. It is now widely acknowledged that the widespread prevalence of weapons as male accoutrements in the Bronze Age has to do with social and political fashions, and may not have anything to do with actual levels of violence. A. Harding has discussed the social persona of the warrior as an emerging figure in the Bronze Age.

We thank the reviewer for this comment. A sentence has been added in Introduction to emphasize the lack of correlation between levels of violence and the quantity of weapons found in Bronze Age burials (l.39-42):

“Moreover, although a large number of individuals have been found buried with weapons in the archaeological records of the Bronze Age, this does not coincide with an increase in interpersonal violence in either Europe or the Middle East, suggesting that the idealisation of the social persona of the warrior may not reflect an intensification of warfare [19–21].”

Indeed, according to Fontijn¹³ which is cited by Harding¹⁴, the increase of weapons in the archeological record during the Bronze Age does not mirror an increase in warfare but instead an “ideology of martiality”. He studied weapons found in the Netherlands and Belgium and found that their manufacture was not suitable for use in battle and their distribution, mainly close to rivers, suggested a more ritualistic use. We have added another reference¹² showing low levels of violence in the Bronze Age compared to earlier or later periods.

- P. 3 lines 65-75. It might also be worth discussing the “stochastic loss of lineages” here; it seems to be implied but not actually mentioned. In any situation tracing lineages of any kind over time from a putative Time 0, some of them will go extinct and others will expand, purely due to random stochastic factors.

We fully agree that some descent groups will expand while others will go extinct even if the variance in reproductive success between groups is set to zero. This stochastic loss is expected to generate a level of drift on the Y chromosome similar to that observed in a patrilocal but not patrilineal system (with the same number of villages and individuals), where Y chromosome lineages are also lost by drift. Therefore, we expect a similar evolution of Y chromosome diversity under scenarios 2a and 2c (patrilocal and patrilineal system with no variance in reproductive success between groups and no violence) as under scenario 1 (patrilocal and bilateral system with no violence) (Figure 2). However, adding variance in reproductive success between patrilineal groups is expected to increase descent group loss and to have a greater effect on Y-chromosome genetic diversity than if there is no variance in reproductive success between groups. We added I.393-401:

“Note that without introducing variance in reproductive success between patrilineal groups, there is no difference in the impact of lineal fission versus random fission on Y-chromosome diversity. In both cases, the drop in Y-chromosome diversity is very limited (Table 2). This can be explained by the reduced number of fission and extinction events in scenarios without variance in reproductive success compared to scenarios with variance in reproductive success between groups (around 25 extinction events over 100 generations in scenarios without variance and without violence compared to around 100 extinctions over 100 generations in scenarios with variance and without violence (Supplementary Fig. 18-25).“

- P. 4 lines 118-124. This could be described a bit more clearly.
 - o (i) if you start with 1500 individuals and have 1% growth per generation over 2500 generations, you end up with considerably more than the current population of the earth. So does the population growth start only after the 2500 generations of panmixia? This seems to be implied in the diagram (Figure 1). Even over only 100 generations, 1% growth is still quite high, resulting in more than doubling the population.

The population begins to increase after 20,000 generations of panmixia. The sentence has been clarified to avoid any confusion (I.117-122):

“we developed a socio-demographic model initially including 1,500 individuals, with males carrying a 1 Mb Y chromosome and females a 10 kb mtDNA. The population size is constant for 20,000 generations and individuals reproduce panmictically. Then, 100 generations before

present, the population is divided into 5 villages of equal size and begins to grow exponentially (1% per generation [46–48]).”

The figure of 1% was chosen based on previous literature assessing the growth rate of ancient populations based on archaeological remains^{47–49}. We believe that this is a reasonable rate. At t_0 , there are 1,500 individuals in the population, then 50 generations later, there are 2355 individuals, and 100 generations later: 3715 individuals.

o (ii) why do you need to model population growth at all, rather than having the population being in a stable demographic regime? Put in some justification of why the model was done this way.

Since we want to compare our simulations with the Skyline plots presented in Karmin *et al.*¹, we have introduced demographic growth in the population in order to reproduce the female effective size increase over time. A sentence has been added to justify this choice (l.122-123):

“Exponential growth is introduced in the model to mimic the dynamics of female effective size shown in Karmin et al. [1].”

We have also modelled a scenario with a first transition from panmixia to a bilateral system and then a second transition to a patrilineal system, with population growth starting at the time of the first transition. Resulting Skyline plots showed that when population growth is older than the onset of patrilineality, the bottleneck of male effective size occurs at a time when female effective size is high, leading to female-to-male effective size ratios exceeding the value of 17 reported by Karmin *et al.*¹ Population growth and its timing are thus important parameters in explaining the bottleneck in male effective size.

This is discussed l.454-458:

“These scenarios also showed that when population growth is older than the onset of patrilineality, the bottleneck of male effective size occurs at a time when female effective size is high, leading to coalescent-based female-to-male effective size ratios comprised between 18.00 (matrilocal residence) and 19.89 (patrilocal residence), similar to the value of 17 reported by Karmin et al [1].”

- It would be good to have some discussion of timing and speed. From the figures in this manuscript, the bottleneck in male genetic diversity would become most evident 40-60 generations (perhaps 1000-1500 years) after the change in lineality rules. The main difference between the “with violence” and “without violence” scenarios is speed: “with violence” scenarios are faster. I understand that Kamin *et al.* reconstructed both the existence of the bottleneck and the timing of it using modern genetic data, and the timing is entirely based on projected trend rates; it is not calibrated to any actual ancient data points from aDNA and the timing of the transition could be rather approximate. Indeed, in many areas, it corresponds better to the onset of the Neolithic (and perhaps new lineality associated with new property relations) than to the beginning of the Bronze Age. So this could be mentioned, not as a way of distinguishing between the two scenarios using actual currently-existing data, but as a way which could be developed in the future of

distinguishing between them by using aDNA to get estimates of genetic diversity at specific dated moments in the past.

In the new version of the manuscript, we now discuss further the differences in speed and timing between scenarios. In violent scenarios the male effective size estimated from nucleotide diversity decreased faster over time than in non-violent scenarios. However, these speed differences are not seen for male effective size estimated with BEAST. A paragraph has been added in Results to describe this difference in timing between violent and peaceful scenarios (l.223-233):

“We also found that different socio-demographic processes acted at different rates on male effective size. In particular, violence had a rapid effect on genetic diversity: in scenarios with variance in reproductive success between groups, the decline in male effective size is much faster in scenarios with violent competition than in scenarios without violent competition (2f compared to 2e, and 2h compared to 2g in Figure 2a). For example, after 20 generations, the male effective size decreased by a factor of 3.30 in the scenario 2h (with violence) and only by 1.36 in the scenario 2g (without violence). However, in the long run, the effect of variance in reproductive success is greater than the effect of violence, and scenarios with violent competition without additional variance are less efficient at reducing male effective size than scenarios with variance in reproductive success but no violent competition between groups (scenario 2b compared to 2e, and 2d compared to 2g).”

We thank the reviewer for their idea to discriminate scenarios using aDNA. We have added the following lines (l.601-607):

“Computations of π -based effective size indicate that violence impacts genetic diversity in a faster way than the kinship system. This outcome implies that future research conducting ancient DNA diversity measurements at multiple time points within the same archaeological site may provide insights into the pace of male effective size reduction, allowing to distinguish between more violent or more peaceful scenarios at local scales, and lead to a more comprehensive picture of the socio-demographic processes at work in ancient human populations.”

- One additional factor that could be mentioned in discussion: the effect of violence here is based on how many males it removes from the reproducing population. It has nothing to do with who is doing the violence or where it comes from. So in that sense, it would be the same whether increased violence came about from an invading horde of warlike people (as in the most extreme of the “Yamnaya invasion” scenarios) or in more realistic and plausible scenarios of endogenous social change without genetic replacement.

We thank the reviewer for this thoughtful comment. We added it in material and methods (l.710-711):

“Note that the source of violence is not specified in the model. It could come from local neighbours or from belligerent outsiders.”

- Throughout: it would be good to modify the language so that the manuscript does not say that villages or people “exchanged women”; instead it can say that women “moved between villages”. Talking about “the exchange of women” is anthropologically traditional

but it implies that women are passive objects and men or groups as a whole are the active agents in society.

Thank you for this comment. We have changed the text as the reviewer suggested.

Please note: as part of this review, I protest Springer Nature's exclusionary policies of high pricing and restrictive open access for a product which is created through donated labour from authors, reviewers and editors; I am donating my labour in this review out of support for colleagues, including the authors and readers, and fairness demands that this should be passed on throughout the system.

Reviewer #3 (Remarks to the Author):

This paper offers an interesting alternative to the model of Zeng et al. (2018), which was developed in an attempt to explain the data of Karmin et al. (2015) showing a seventeen to one ratio of females to males in post-Neolithic European humans. Zeng et al. incorporated cultural transmission belonging to Y chromosome haplogroups that competed with one another ecologically, resulting in the demise of enough Y-chromosome haplotypes to produce an extremely female-biased population.

Guyon et al. develop a numerical simulation of Y chromosomes and Mitochondria that involve an array of demographic assumptions.

1. The overall population size in the model simulation is 1,500, which is divided into five "villages" of 300 each.

2. In one model there is random reproduction between males and females among the 300. It is not explained how the sex ratio at birth is determined; it is not stated that there are 150 males and 150 females at the start of the simulation in each group.

The sex ratio is balanced in the initial population (750 males and 750 females) and is maintained in the formation of villages and descent groups. In addition, the sex of children is assigned so that the sex ratio is balanced within villages in each generation. We added these precisions in the Methods section I.628-629:

"After 20,000 generations, the population divides into 5 villages, each with the same size ($N_0 = 300$ individuals) and a balanced sex ratio."

and I.671-672:

"When children are born, they are alternately assigned a male and a female sex, so as to maintain a balanced sex ratio in the villages."

The graph below shows measures of the sex ratio after each reproduction event in all replicates of simulation for the scenario 2g:

Generations for which the sex ratio is not exactly 0.5 correspond to cases of odd numbers of individuals in villages.

3. In the second model, each group of 300 is divided into three groups of 100. It is not stated whether there are initially fifty males and fifty females in each descent group.

Indeed, there are initially fifty males and fifty females in each descent group. We added these precisions in the Methods section (l.630-631):

“In the case of patrilineal descent, each village is structured into 3 descent groups of initial size $K = 100$ individuals with a balanced sex ratio.”

4. The growth rates of descent groups are chosen from a normal distribution initially. It is unclear what happens when (and if) a descent group has ninety-nine males and one female, for example.

This situation cannot occur because males can mate with any female in the village and the sex ratio of the village is balanced. In addition, when a descent group splits, the proportion of females going to a new descent group is equal to the proportion of males forming the new descent group. This precision was added in Materials and Methods, l.686-689:

“Once the males are distributed in the two new groups, the females are split between the two groups. For a given group, the proportion of sampled females is equal to the proportion of males sampled from the ancestral group, so that sex-ratio is balanced in the new descent groups.”

The village sex ratio may be slightly skewed after migration of individuals between villages, but is still close to 0.5 as shown in the following figure:

Similarly, post-fission migration of descent groups may slightly unbalance the sex ratio, but it is always comprised between 0.498 and 0.510, as shown below:

Post-fission migration cannot lead to a very unbalanced sex ratio since descent groups move to a village where another descent group has moved when possible.

It is stated that descent groups “undergo fusion” when their size exceeds 100. But they start at 100?

Groups start with 100 individuals, and split when they reach the fission threshold (100, 150, 200) and when at least 3 generations have passed since the last split (or since the initial formation of descent groups). In the new version, we have decided to show the results for a fission threshold of 150 individuals instead of 100 in the main figures, because this is the intermediate value between 100 and 200. In the case when the fission threshold is 100, groups initially formed at t_0 must wait 3 generations before they

split. In the meantime, depending on their growth rates, some groups will grow above 100 and others will fall below 100.

If a descent group goes extinct, what happens to the system? Can all descent groups go extinct? Do males and females have the same birth rates?

The size of a village is controlled over time (it increases exponentially, 1% per generation). What we previously called “growth rates” of descent groups were actually “relative fitnesses” used to compute a probability of having children. Therefore we replaced “growth rates” by “relative fitnesses” in the text. The relative fitness of each group is normalized according to relative fitnesses of other groups from the same village (see equation 1 in Material and Methods). So when a group goes extinct, other groups in the village grow more than the mean growth rate of 1%. Hence, it is impossible that all groups go extinct. All males and females from the same group have the same probability of having children (computed from the relative fitness of the group).

5. Female migration. If the female migration rate between villages is 10% (line 400), does this mean on average fifteen females from each village move to a different village, and are these females distributed equally among the five descent groups?

Yes it means that for a village with 150 females, 15 of them migrate to another village. The village, as well as the descent group to which they migrate, is drawn randomly for each female.

6. There have to be detailed assumptions on reproduction between males and local females and between males and immigrant females.

There is no difference in reproduction between local and immigrant females. Migration happens before mating and mating pairs are randomly drawn in the village.

What does minimizing “the number of individuals to remain single” entail? “Descent groups display different growth rates.” There are fifteen descent groups. Once an r_i is chosen, how are the mating pairs chosen?

When forming mating pairs, we randomly draw a male and a female from different descent groups. However, if at some point there are only males and females from the same group left, we draw pairs in the same group. As the sex ratio is close to 0.5, few individuals will remain single. We have modified the description in Results and Methods to clarify this point (l.646-649):

“Deviations from strict descent group exogamy may occur when a male and a female from different descent groups cannot be mated. In this case, couples are formed by randomly selecting a male and a female from the same descent group, to minimise the number of individuals that remain single.”

At each generation, the offspring population is simulated at the village level. Each child is sequentially assigned to a randomly selected mating pair in the village, such that the probability P_j that the father of the child belongs to the descent group j is given by:

$$P_j = \frac{2 \times M_j \times \exp(r_j)}{\sum_{i=1}^n 2 \times M_i \times \exp(r_i)}$$

where r_j is the relative fitness of the descent group j , M_j is the number of males in the descent group j and n is the number of descent groups in the village (see Materials and Methods, 1.650-657).

7. How many descent groups are eventually seen? If the descent groups get too big, can the sex ratio in a split-off group become extreme?

The number of descent groups over time in the population (averaged over replicates) is shown in Supp Fig 18 to 25. Some scenarios show up to 60 descent groups, i.e. an average of up to 12 descent groups per village.

When a split occurs, a proportion p of the males form a new group. Then, a number ($p \times nb$ of females in the group) of females are drawn to join them in the new group. Therefore, the sex ratio can be slightly unbalanced but not extreme as shown in the preceding graph.

All of the above issues pertain only to the generation-to-generation samplings, not to the estimates of N_e^{Y} , N_e^{mt} obtained from π and BSP. These also involve assumptions on the mutation rates for Y and mt, in π and BSP.

Indeed, estimates of N_e from π are obtained by dividing π by the mutation rate used in the simulation. Similarly, the mutation rate is a parameter used by BEAST for the demographic inference. When estimating N_e from simulated data using these two methods, we used the same mutation rates for the Y-chromosome and mtDNA as the ones used in the simulations, so these assumed mutation rates do not impact our estimates. This is not the case for real data analysis, like the ones performed by Karmin *et al.*¹ and Batini *et al.*^{34,45}, in which assumed mutation rates can affect the inference of the timing of the bottleneck. For example, Batini *et al.*⁴⁵, who used a faster mutation rate for the Y-chromosome than Karmin *et al.*¹, estimated a more recent bottleneck of male effective size.

Assuming the same numbers were used, there is an enormous discrepancy in the male N_e and the female-to-male effective sizes using the two methods. This difference is not discussed.

Indeed, the same numbers were used in the two methods: there are the same numbers used in the simulations. The differences are due to the fact that Bayesian skyline plots infer instantaneous effective sizes of the population's past from current individuals by measuring coalescence rates along coalescent trees, while nucleotide diversity (π) estimates the effective size of the population by integrating its past dynamics, which is equivalent to computing the average of the coalescence times of all pairs of individuals in the sample. A paragraph has been added to the Discussion to address this concern (1.358-367):

“The two methods used to compute effective sizes showed similar relative effects of the different parameters in the model, but reported different bottleneck amplitudes for the male effective size. In addition, coalescent-based male effective sizes were larger after 100 generations of patriliney while π -based male effective size were lower, suggesting that the coalescent-based method captures more of the growth signal than the π -based method. These discrepancies can be explained by the fact that Bayesian skyline plots infer instantaneous effective sizes of the population’s past from current individuals by measuring coalescence rates along coalescent trees. In contrast, nucleotide diversity (π) estimates the effective size of the population by integrating its past dynamics, which is equivalent to computing the average of the coalescence times of all pairs of individuals in the sample.”

In addition, the value “17” from Karmin et al. does not appear in Table 2, and in Figure 5 the average is 8.86. How does this compare with Zeng et al.? The authors call Zeng et al.’s model “Violence” but do not say how that model is included in the simulation to produce Figure 4. Are the cultural groups of Zeng et al. equivalent to the descent groups here?

In Table 2, the value of “17” from Karmin *et al.*¹ is written in the column “Maximum coalescent-based female-to-male effective size ratio”. The average we obtained using skyline plots with the scenario 2g, in the absence of male and group migration, is 7.63. When we take into account post-fission migration, the value exceeds 20 (Supplementary Table 5). When considering an initial shift to bilateral descent, followed by a shift to patriliney (with the settings of scenario 2g), the value is above 18 (Table 2).

These values are not comparable with the value obtained by Zeng *et al.*¹⁵ who did not perform skyline plots. Zeng *et al.*¹⁵ measured the Y-chromosome haplogroup diversity with a metric (Shannon entropy) that is not comparable to coalescent-based male effective size. It is also difficult to compare it with our estimates of nucleotide diversity (π). However, the Y chromosome Shannon entropy is decreased by 4 when running Zeng *et al.*’s model¹⁵, which is in the same magnitude order of the reduction by a factor of 2.14 of the π -based male effective size we observed for scenario 2b with no male and group migration (the most similar to Zeng *et al.*’s model¹⁵).

We specified in line 184 that the scenario 2b is similar to the one proposed by Zeng *et al.*¹⁵ In particular violence is modelled in the same way (as explained in materials and methods). “Cultural groups” / “kin groups” from Zeng *et al.*¹⁵ are indeed equivalent to “descent groups” in our study as explained in Supplementary Table 3:

“Patrilineal kin groups are called descent groups in our study, following Fox’s terminology [10], and cultural groups or kin groups in Zeng et al.’s study [9].”

and as explained in the introduction lines 19-22:

“Thus, Zeng et al. [9] simulated a human population structured into competing patrilineal kin groups with each other. In this model, kin groups - called descent groups in our article, following Fox’s terminology [10] - are genetically homogeneous for the Y-chromosome because men in the same group share a recent common paternal ancestor.”

I think the paper loses its impact by not having a table of parameters and their values. It

is too important to be left to a supplement. I believe more care in the writing could produce an acceptable manuscript. It is a very interesting topic.

We thank the reviewer for this suggestion. The table was displaced in Material and Methods after l.716.

Minor points

Line 21. Our paper. Is this the present manuscript? It refers to Fox which is (10).

Yes it is the present manuscript. We modified the text to avoid this confusion. The reference to Fox⁵⁰ is there to justify the usage of the expression “descent group” instead of “kin group”.

Lines 50-51. “This was reported” — not “Such observation”.

Corrected.

Line 57 and elsewhere. “affiliated with” not “to”

Corrected.

Lines 73, 81. “This variance” — not “Such variance”

Corrected

Line 88. Define “segmentary patrilineal system”

A definition is now given l.83-86:

“Secondly, some patrilineal systems, known as segmentary patrilineal systems, are characterised not only by patrilineal descent but also by the segmentation (also called fission) of descent groups when they become too large [10, 40]. These systems account for 14% of patrilineal systems according to Aberle [41].”

Lines 118-124. The mating system is not clear. How are mates chosen?

We have rephrase this paragraph (l.133-143) and added a few lines to explain better how mates are chosen:

“Mating pairs are formed by randomly drawing individuals from two different descent groups, unless all remaining single individuals are from the same descent group. In this case, they are randomly drawn from the same descent group. While the size of villages increases exponentially with a fixed rate (1%), descent groups have different growth rates (see Equation 1, Materials and Methods). The growth rates of descent groups depend on their relative fitnesses that are initially drawn from a normal distribution with mean $r = 0.01$ and variance σ^2 that can be zero (i.e., no variance, all descent groups have the same growth rate), low ($\sigma^2 = 0.05$), intermediate ($\sigma^2 = 0.1$) or high ($\sigma^2 = 0.2$). These values have been chosen so that the growth and extinction rates of descent groups are consistent with those found in the literature focusing on patrilineal populations without mentioning warfare between descent groups (Supplementary Table 1, 2).”

What happens if the sex-ratio is not even?

If the sex ratio deviates from 0.5, some individuals remain single. However, the sex ratio is controlled in the simulations and is always close to 0.5 so there are few single individuals at each generation.

The graphs go back 125 generations. Why?

In total, 20,100 generations are simulated for scenarios 1 and 2, and 20,200 for scenario 3. We are interested in the 100 last generations, corresponding to the period when kinship systems of interest are at work. When producing Skyline Plots with BEAST, it is possible that the skyline starts before the last 100 generations. Here we chose to plot them from -125 generations so that the transition between panmixia and the new kinship system is visible in the BSPs.

Line 134 and elsewhere. Use “exceeds” everywhere instead of “overpasses”

Corrected

Line 192. “dropped by a factor of 27.6..”

Corrected

Figure 5, legend. “...Y chromosome and mtDNA effective population sizes under...”

Corrected

References

1. Karmin, M. & al. A recent bottleneck of Y chromosome diversity coincides with a global change in culture. *Genome Res.* **25**, 459–466 (2015).
2. Marlowe, F. W. Marital Residence among Foragers. *Curr. Anthropol.* **45**, 277–284 (2004).
3. Andersen, K. K. *et al.* High-resolution record of Northern Hemisphere climate extending into the last interglacial period. *Nature* **431**, 147–151 (2004).
4. Bowles, S. Did Warfare Among Ancestral Hunter-Gatherers Affect the Evolution of Human Social Behaviors? *Science* **324**, 1293–1298 (2009).
5. Crevecoeur, I., Dias-Meirinho, M.-H., Zazzo, A., Antoine, D. & Bon, F. New insights on interpersonal violence in the Late Pleistocene based on the Nile valley cemetery of Jebel Sahaba. *Sci. Rep.* **11**, 9991 (2021).

6. Hames, R. Pacifying Hunter-Gatherers. *Hum. Nat.* **30**, 155–175 (2019).
7. Fry, D. P. & Söderberg, P. Lethal Aggression in Mobile Forager Bands and Implications for the Origins of War. *Science* **341**, 270–273 (2013).
8. Wrangham, R. W. & Glowacki, L. Intergroup Aggression in Chimpanzees and War in Nomadic Hunter-Gatherers. *Hum. Nat.* **23**, 5–29 (2012).
9. Pinker, S. *The Better Angels of Our Nature: The Decline of Violence In History And Its Causes*. (Penguin UK, 2011).
10. Gómez, J. M., Verdú, M., González-Megías, A. & Méndez, M. The phylogenetic roots of human lethal violence. *Nature* **538**, 233–237 (2016).
11. Meyer, C. *et al.* Early Neolithic executions indicated by clustered cranial trauma in the mass grave of Halberstadt. *Nat. Commun.* **9**, 2472 (2018).
12. Baten, J., Benati, G. & Sołtysiak, A. Violence trends in the ancient Middle East between 12,000 and 400 bce. *Nat. Hum. Behav.* (2023) doi:10.1038/s41562-023-01700-y.
13. Fontijn, D. Giving up weapons. in 145–154 (Oxford: Archaeopress, 2005).
14. Harding, A. Bronze Age Encounters: Violent or Peaceful? *Warf. Violence Slavery Prehistory* 145–154 (2018).
15. Zeng, T. C., Aw, A. J. & Feldman, M. W. Cultural hitchhiking and competition between patrilineal kin groups explain the post-Neolithic Y-chromosome bottleneck. *Nat. Commun.* **9**, 2077 (2018).
16. Aberle, D. F. Matrilineal descent in cross-cultural perspective. 655–727 (1961).
17. Holden, C. J. & Mace, R. Spread of cattle led to the loss of matrilineal descent in Africa: a coevolutionary analysis. *Proc. R. Soc. Lond. B Biol. Sci.* **270**, 2425–2433 (2003).
18. Shenk, M. K., Begley, R. O., Nolin, D. A. & Swiatek, A. When does matriline fail? The frequencies and causes of transitions to and from matriline estimated from a de novo coding of a cross-cultural sample. *Philos. Trans. R. Soc. B Biol. Sci.* **374**, 20190006 (2019).
19. Murdock, G. P. *Social Structure*. xvii, 387 (Macmillan, Oxford, England, 1949).
20. Engels, F. *Origin of the Family, Private Property and the State*. vol. 3 (Hottingen-Zurich,

- 1884).
21. Hartung, J. *et al.* On Natural Selection and the Inheritance of Wealth [and Comments and Reply]. *Curr. Anthropol.* **17**, 607–622 (1976).
 22. Schroeder, H. *et al.* Unraveling ancestry, kinship, and violence in a Late Neolithic mass grave. *Proc. Natl. Acad. Sci.* **116**, 10705–10710 (2019).
 23. Sánchez-Quinto, F. *et al.* Megalithic tombs in western and northern Neolithic Europe were linked to a kindred society. *Proc. Natl. Acad. Sci.* **116**, 9469–9474 (2019).
 24. Mitnik, A. *et al.* Kinship-based social inequality in Bronze Age Europe. *Science* **366**, 731–734 (2019).
 25. Seguin-Orlando, A. *et al.* Heterogeneous Hunter-Gatherer and Steppe-Related Ancestries in Late Neolithic and Bell Beaker Genomes from Present-Day France. *Curr. Biol. CB* **31**, 1072-1083.e10 (2021).
 26. Fowler, C. *et al.* A high-resolution picture of kinship practices in an Early Neolithic tomb. *Nature* (2021) doi:10.1038/s41586-021-04241-4.
 27. Rivollat, M. *et al.* Ancient DNA gives new insights into a Norman Neolithic monumental cemetery dedicated to male elites. *Proc. Natl. Acad. Sci. U. S. A.* **119**, e2120786119 (2022).
 28. Rivollat, M. *et al.* Extensive pedigrees reveal the social organization of a Neolithic community. *Nature* **620**, 600–606 (2023).
 29. Blöcher, J. *et al.* Descent, marriage, and residence practices of a 3,800-year-old pastoral community in Central Eurasia. *Proc. Natl. Acad. Sci.* **120**, e2303574120 (2023).
 30. Žegarac, A. *et al.* Ancient genomes provide insights into family structure and the heredity of social status in the early Bronze Age of southeastern Europe. *Sci. Rep.* **11**, 10072 (2021).
 31. Villalba-Mouco, V. *et al.* Kinship practices in the early state El Argar society from Bronze Age Iberia. *Sci. Rep.* **12**, 22415 (2022).
 32. Service, E. R. *The Hunters*. (Prentice-Hall, 1966).
 33. Brewer, D. D. *A Systematic Review of Post-Marital Residence Patterns in Prehistoric*

Hunter-Gatherers. <http://biorxiv.org/lookup/doi/10.1101/057059> (2016)

doi:10.1101/057059.

34. Batini, C. *et al.* Population resequencing of European mitochondrial genomes highlights sex-bias in Bronze Age demographic expansions. *Sci. Rep.* **7**, 12086 (2017).
35. Balanovsky, O. *et al.* Genetic differentiation between upland and lowland populations shapes the Y-chromosomal landscape of West Asia. *Hum. Genet.* **136**, 437–450 (2017).
36. Scorrano, G., Yediay, F. E., Pinotti, T., Feizabadifarahani, M. & Kristiansen, K. The genetic and cultural impact of the Steppe migration into Europe. *Ann. Hum. Biol.* **48**, 223–233 (2021).
37. Ly, G. *et al.* Residence rule flexibility and descent groups dynamics shape uniparental genetic diversities in South East Asia. *Am. J. Phys. Anthropol.* **165**, 480–491 (2018).
38. Hamilton, G., Stoneking, M. & Excoffier, L. Molecular analysis reveals tighter social regulation of immigration in patrilocal populations than in matrilineal populations. *Proc. Natl. Acad. Sci.* **102**, 7476–7480 (2005).
39. Forde, C. D. Fission and Accretion in the Patrilineal Clans of a Semi-Bantu Community in Southern Nigeria. *J. R. Anthropol. Inst. G. B. Irel.* **68**, 311–338 (1938).
40. Rapper, G. de. Blood and Seed, Trunk and Hearth: Kinship and Common Origin in Southern Albania. in 79 (LIT Verlag, 2012).
41. Encyclopedia of Sex and Gender: Men and Women in the World's Cultures. *Ref. Rev.* **18**, 19–20 (2004).
42. Mulder, M. B. On Cultural and Reproductive Success: Kipsigis Evidence. *Am. Anthropol.* **89**, 617–634 (1987).
43. Mulder, M. B. Kipsigis women's preferences for wealthy men: evidence for female choice in mammals? *Behav. Ecol. Sociobiol.* **27**, 255–264 (1990).
44. Ross, C. T. *et al.* Greater wealth inequality, less polygyny: rethinking the polygyny threshold model. *J. R. Soc. Interface* **15**, 20180035 (2018).
45. Batini, C. *et al.* Large-scale recent expansion of European patrilineages shown by population resequencing. *Nat. Commun.* **6**, 7152 (2015).

46. Skov, L. *et al.* Genetic insights into the social organization of Neanderthals. *Nature* **610**, 519–525 (2022).
47. Carneiro, R. L. & Hilse, D. F. On Determining the Probable Rate of Population Growth During the Neolithic. *Am. Anthropol.* **68**, 177–181 (1966).
48. Hassan, F. A. & Sengel, R. A. On Mechanisms of Population Growth During the Neolithic. *Curr. Anthropol.* **14**, 535–542 (1973).
49. Zahid, H. J., Robinson, E. & Kelly, R. L. Agriculture, population growth, and statistical analysis of the radiocarbon record. *Proc. Natl. Acad. Sci.* **113**, 931–935 (2016).
50. Fox, R. *Kinship and Marriage: An Anthropological Perspective*. (Cambridge Univ. Press, Cambridge, 2006).

Reviewers' Comments:

Reviewer #1:

Remarks to the Author:

I thank the authors for their responses to the concerns raised in my initial review. I think this paper is generally improved by the analyses of additional scenarios, but it has become much more difficult to read with the addition of many long-winded paragraphs.

The authors should avoid simply listing the values they have obtained from their simulations in the results section, especially because a lot of this information is also in Table 2. This bogs down the main text with very specific numbers, when a qualitative summary of the observed differences between scenarios would be sufficient and also more useful (lines 340-346 provide a good example of this, but this tendency is present throughout the paper). I don't put much stock in the specific values obtained from this highly simplified set of models. Rather, in my opinion, the usefulness of this framework comes mainly from the ability to explore relative differences between demographic scenarios.

I feel that the paper would benefit from more structure or an overarching hypothesis that more explicitly connected to human history. The link between shifts in kinship systems and subsistence strategies should be emphasized much more than it currently is if that is in fact the authors' primary argument.

Wording issues

Abstract line 1: 'Studies have found'

Abstract line 12: 'without resorting to/invoking the violence hypothesis'

Line 6: 'by a study by Poznik et al.'

Line 44: 'genetic diversity'

Line 148: 'the variance is set to the value'

Line 154: 'subdivided' – not subdivised

Line 198-218: The results are not reported using consistent language to describe the magnitude of the declines observed under different scenarios. This is confusing for a reader to parse. The authors should stick with a single phrasing whenever they are reporting this kind of result (not just in this section), even if repetitive (e.g. 'the effective population size of males decreased by a factor of X').

Line 287: those not the one.

Line 290: change in not evolution of – same in Figure 6 title

Lines 302-310: Switches from putting the scenario number in brackets to putting it in the main text. The former is much easier to follow, but it should at least be consistent.

Line 315: Don't refer to the scenarios by their shorthand labels. Use the full description, i.e. which parameters differ between scenarios.

Lines 351-352: 'on the one hand' and 'on the other hand' are unnecessary.

General note on the discussion: sections that are simple summaries should be cut out

Line 376-377: These results are hardly comparable? Then why are they being compared?

Line 514-515: In East Africa or the Arabian Peninsula, it was shown that mobile pastoralism predated agriculture. Why was this sentence included?

Lines 540-544: This section starts by stating that polygyny was not included in the model, but then goes on to describe a scenario where it was included.

Tables and figures

Figure 2: It isn't possible to clearly see all scenarios in Figure 2, as many of the points/lines overlap with each other (especially in b – it is essentially impossible to discern the differences between scenarios). It seems like the authors have tried to address this by staggering the points along the x-axis, but I think this needs to be exaggerated. The same should be done for Figures 3-6 as well for consistency.

Table 2 – NA instead of unknown. On average instead of in average. The scenario parameters should be listed here instead of the numbers. Readers should not have to go back to a figure legend to interpret this table.

Reviewer #3:

Remarks to the Author:

The paper has been nicely revised with all of the reviewers' points satisfactorily addressed.

REVIEWERS' COMMENTS

Reviewer #1 (Remarks to the Author):

I thank the authors for their responses to the concerns raised in my initial review. I think this paper is generally improved by the analyses of additional scenarios, but it has become much more difficult to read with the addition of many long-winded paragraphs.

We thank the reviewer for their helpful comments. For a more fluid reading, the results and discussion have been reformulated as explained in more details below.

The authors should avoid simply listing the values they have obtained from their simulations in the results section, especially because a lot of this information is also in Table 2. This bogs down the main text with very specific numbers, when a qualitative summary of the observed differences between scenarios would be sufficient and also more useful (lines 340-346 provide a good example of this, but this tendency is present throughout the paper).

In the results section, most of the listed values have been removed, and the following paragraphs have been reformulated for ease of understanding:

- lines 229-237:

“Table 2 and Figure 2 show that the male effective population size decreased by the same amount (below 2) after 100 generations when there was a shift to bilateral descent and patrilocal residence (scenario 1) and when there was a shift to patrilineal descent and patrilocal residence with random fission, no variance in reproductive success and no violence (scenario 2a: basic patrilineal scenario) (Figure 2a). Adding violent intergroup competition (scenario 2b, the most similar to that proposed by Zeng et al.[9]), variance in reproductive success between patrilineal descent groups ($\sigma^2= 0.1$) (scenario 2e), or both violent intergroup competition and variance in reproductive success (scenario 2f) to the basic patrilineal scenario reduced the male effective population size by 2 to 4, 100 generations after t_0 .”

- lines 327-332:

“In the scenario with patrilineal descent, variance in reproductive success between groups, and no violence, taking into account lineal rather than random fission resulted in a greater reduction in the male effective population size (scenarios 2e and 2g in Table 2). In scenarios with patrilineal descent and lineal fission, the introduction of variance in reproductive success (scenario 2g) resulted in a greater reduction in the male effective population size compared to the introduction of violence (scenario 2d).”

- lines 343-349:

“These ratios were lower when considering 2% intervillage male migration but no post-fission migration (Supplementary Fig. 12). However, the ratios increased substantially when post-fission migration was taken into account, with or without 2% intervillage male migration (Supplementary Fig. 13, 14). Considering both types of migration, the female-to-male effective population size ratio reached 14.20 in the scenario with patrilineal descent, lineal fission, variance and no violence (Supplementary Fig. 14).”

- lines 360-367:

“Figure 6 shows that the bottleneck in male effective population size is maintained even when patrilineality is introduced after a phase with a bilateral descent system, regardless of the residence

rule. Indeed, the male effective population size reduction factor reached similar values (around 3) with an initial bilateral and patrilocal phase, with an initial bilateral and multilocal phase, and with an initial bilateral and matrilineal phase. As the effective population size of females increases earlier than in the previous scenarios, it is higher when the bottleneck in male effective population size occurs. This results in higher peaks in the female-to-male effective population size ratio, which reached values around 19 in all three scenarios.”

In the Discussion section, some values correspond to information that is not presented in the Results section but is necessary to fuel the discussion. For instance, the timing of the bottleneck or the impact of polygyny are discussed and require citation of relevant values for a better understanding.

I don't put much stock in the specific values obtained from this highly simplified set of models. Rather, in my opinion, the usefulness of this framework comes mainly from the ability to explore relative differences between demographic scenarios.

Most values have been withdrawn from the main text. Some values have been highlighted to show that the magnitude of the bottleneck reported by Karmin *et al.* can be achieved with our model. However, we now focus on the differences between the scenarios rather than the absolute values.

eg. lines 328-329 : “taking into account lineal rather than random fission, resulted in a greater reduction in the male effective population size”

I feel that the paper would benefit from more structure or an overarching hypothesis that more explicitly connected to human history. The link between shifts in kinship systems and subsistence strategies should be emphasized much more than it currently is if that is in fact the authors' primary argument.

To better emphasise the main hypothesis of the paper, we moved the paragraph explaining the relationship between subsistence strategies and kinship systems from discussion to introduction (l.66-85). We also added a sentence at the beginning of this paragraph to formulate an overarching hypothesis:

“Here we test the hypothesis that the Y-chromosome bottleneck is the result of a global shift towards patrilineal systems, associated with the transition to new subsistence systems on all continents over the past 12,000 years.”

And the sentences l. 121-126 were modified as follows:

“In this study, we undertake a modelling approach to test our hypothesis that a transition to patrilineal organisations, linked to a worldwide change in subsistence strategies, may have triggered an important loss of Y-chromosome diversity and may be sufficient to explain the post-Neolithic Y-chromosome bottleneck reported by Karmin *et al.* [1] without requiring a violent scenario. By simulating different socio-demographic models, we assess the effect of patrilocal residence and patrilineal descent on uniparental genetic diversity.”

Wording issues

Abstract line 1: ‘Studies have found’ corrected

Abstract line 12: 'without resorting to/invoking the violence hypothesis' corrected

Line 6: 'by a study by Poznik et al.' corrected

Line 44: 'genetic diversity' corrected

Line 148: 'the variance is set to the value' corrected

Line 154: 'subdivided' – not subdivised corrected

Line 198-218: The results are not reported using consistent language to describe the magnitude of the declines observed under different scenarios. This is confusing for a reader to parse. The authors should stick with a single phrasing whenever they are reporting this kind of result (not just in this section), even if repetitive (e.g. 'the effective population size of males decreased by a factor of X').

A general effort in homogenizing the formulation was made.

lines 229-247: scenarios are systematically compared with the basic patrilineal scenario (2a) using the formulation: "Adding A to the basic patrilineal scenario reduced the male effective population size by x" or "Taking into account A rather than B in the basic patrilineal scenario reduced the male effective population size by x" (where A and B can be: lineal fission, random fission, variance in reproductive success or violent intergroup competition).

lines 318-340: we use the formulation "In scenario S, adding A resulted in a greater reduction in the male effective population size" or "In scenario S, taking into account A rather than B resulted in a greater reduction in the male effective population size" (where A and B can be: lineal fission, random fission, variance in reproductive success or violent intergroup competition).

Line 287: those not the one. corrected

Line 290: change in not evolution of – same in Figure 6 title

We have applied this suggestion to the entire text.

Lines 302-310: Switches from putting the scenario number in brackets to putting it in the main text. The former is much easier to follow, but it should at least be consistent.

We have chosen to put all the scenario numbers in brackets.

Line 315: Don't refer to the scenarios by their shorthand labels. Use the full description, i.e. which parameters differ between scenarios.

The sentence was changed according to this suggestion (l.336-340):

"Note however that relatively, the values of both approaches are consistent, with higher reduction factors for scenarios with patrilineal descent, variance in reproductive success between groups, and lineal fission (2g and 2h), and lower reduction factors for scenarios with bilateral descent (1) or patrilineal descent with no variance and no violence (2a and 2c)."

Lines 351-352: 'on the one hand' and 'on the other hand' are unnecessary. corrected

General note on the discussion: sections that are simple summaries should be cut out

To address this concern, we have removed the third paragraph of the discussion, as well as some sentences that only summarised the results.

Line 376-377: These results are hardly comparable? Then why are they being compared?

The scenario 2b was built to resemble Zeng *et al.*'s model with a more realistic mutation model and population structure. It seemed important to us to justify why there are discrepancies between our model and Zeng *et al.*'s model since they are similar in many respects (see Supplementary Table 3). However, we agree that the sentence is confusing and decided to remove it.

Line 514-515: In East Africa or the Arabian Peninsula, it was shown that mobile pastoralism predated agriculture. Why was this sentence included?

This sentence was added to show that pastoralism emerged at several places in the world. To emphasise our argument, we changed the sentence as follows (l.493-495):

“In East Africa or the Arabian Peninsula, mobile pastoralism emerged between 7,000 and 5,000 BP (Harrower 2010)”

Lines 540-544: This section starts by stating that polygyny was not included in the model, but then goes on to describe a scenario where it was included.

The sentence was changed by removing “Although we have not included them in our model” (l.520) to avoid this confusion.

Tables and figures

Figure 2: It isn't possible to clearly see all scenarios in Figure 2, as many of the points/lines overlap with each other (especially in b – it is essentially impossible to discern the differences between scenarios). It seems like the authors have tried to address this by staggering the points along the x-axis, but I think this needs to be exaggerated. The same should be done for Figures 3-6 as well for consistency.

The distance between points corresponding to different scenarios was increased in all figures.

Table 2 – NA instead of unknown. On average instead of in average. corrected

The scenario parameters should be listed here instead of the numbers. Readers should not have to go back to a figure legend to interpret this table.

Parameters are now written in the legend of the table.

Reviewer #3 (Remarks to the Author):

The paper has been nicely revised with all of the reviewers' points satisfactorily addressed.

We thank the reviewer for their positive feedback.